# Targeting the JAK/STAT Signaling Pathway Using Phytocompounds for Cancer Prevention and Therapy

**DOI:** 10.3390/cells9061451

**Published:** 2020-06-11

**Authors:** Sankhadip Bose, Sabyasachi Banerjee, Arijit Mondal, Utsab Chakraborty, Joshua Pumarol, Courtney R. Croley, Anupam Bishayee

**Affiliations:** 1Department of Pharmacognosy, Bengal School of Technology, Chuchura 712 102, India; sankha.bose@gmail.com; 2Department of Phytochemistry, Gupta College of Technological Sciences, Asansol 713 301, India; sabyasachibanerjee04@gmail.com (S.B.); utsabcky@gmail.com (U.C.); 3Department of Pharmaceutical Chemistry, Bengal College of Pharmaceutical Technology, Dubrajpur 731 123, India; 4Lake Erie College of Osteopathic Medicine, Bradenton, FL 34211, USA; JPumarol36499@med.lecom.edu (J.P.); CCroley48578@med.lecom.edu (C.R.C.)

**Keywords:** cancer, Janus kinase, signal transducer and activator of transcription, natural compounds, targeted therapy

## Abstract

Cancer is a prevalent cause of mortality around the world. Aberrated activation of Janus kinase (JAK)/signal transducer and activator of transcription (STAT) signaling pathway promotes tumorigenesis. Natural agents, including phytochemicals, exhibit potent anticancer activities via various mechanisms. However, the therapeutic potency of phytoconstituents as inhibitors of JAK/STAT signaling against cancer has only come into focus in recent days. The current review highlights phytochemicals that can suppress the JAK/STAT pathway in order to impede cancer cell growth. Various databases, such as PubMed, ScienceDirect, Web of Science, SpringerLink, Scopus, and Google Scholar, were searched using relevant keywords. Once the authors were in agreement regarding the suitability of a study, a full-length form of the relevant article was obtained, and the information was gathered and cited. All the complete articles that were incorporated after the literature collection rejection criteria were applied were perused in-depth and material was extracted based on the importance, relevance, and advancement of the apprehending of the JAK/STAT pathway and their relation to phytochemicals. Based on the critical and comprehensive analysis of literature presented in this review, phytochemicals from diverse plant origins exert therapeutic and cancer preventive effects, at least in part, through regulation of the JAK/STAT pathway. Nevertheless, more preclinical and clinical research is necessary to completely comprehend the capability of modulating JAK/STAT signaling to achieve efficient cancer control and treatment.

## 1. Introduction

Cancer, a significant and serious medical issue, is driving mortality throughout the globe [1]. It is distinguished by uncontrolled and unscheduled cell multiplication. Tumors associated with viral or bacterial infection and genetic mutations are known to influence the cancer growth rates [2]. Although cancer is thought to be preventable, there are several factors that will increase the risk for developing it. Various risk factors, such as smoking and chewing tobacco, alcohol, obesity, chronic inflammation, age, ethnicity, and geographical location, are the major determinants for developing cancer [3]. Emerging research suggests that the majority of cancers are brought about by malfunction of numerous protein coded genes, such as antiapoptotic proteins, growth factors, receptors of growth factor, tumor suppressors, and transcription factors. These factors constitute the target for cancer prevention and treatment. 

When tumors are formed, their abnormal cells obtain at least 10 common characteristics that describe the malignant tumor cell’s complexity, including proliferation, resistance to growth suppressors, avoidance of programmed cell death (apoptosis), unlimited ability to replicate, growth of fresh blood vessels, tissue invasion with risk of metastatic growth, genetic instability and mutation of high frequency, tumor-driven inflammation, altered metabolism, and compromised immunological surveillance [4,5]. During tumor advancement and malignant conversion, abnormal cells evade the mechanisms of host defense. Targeting and inhibiting the previously mentioned characteristics has been perceived as potential approaches to cancer treatment. Apart from conventional treatment, an additional area of interest among researchers is the utilization of alternative therapy, particularly using dietary sources to regulate these key mechanisms. 

The idea of chemoprevention is receiving extensive awareness because it is a cost-effective option for cancer control when compared to conventional chemotherapy [6,7]. Cancer chemoprevention by utilizing natural and dietary compounds, particularly phytoconstituents, minerals, and vitamins, in different in vivo and in vitro conditions, has produced empowering results against several cancers [8,9,10,11,12]. Herbal medicines are emerging as new and innovative substances with astounding pharmaceutical potential that can be utilized to abate human ailments, including cancer [13,14]. No conclusive evidence has been found that herbal medicines can cure or treat cancer by themselves. Nevertheless, certain plant extracts, fractions, and pure compounds have been found to possess anticancer properties, and are used to develop chemotherapy medicines. For example, vincristine and vinblastine, the active constituents of *Catharanthus roseus*, and taxanes present in Pacific yew tree, are approved chemotherapy drugs. 

There is a correlation between dietary habits and cancer development of certain cancers. Food contains different bioactive compounds that improve health and counteract cancer and other chronic diseases. These bioactive elements are found mostly in plants [15,16,17,18,19,20]. Various findings have led to the identification and development of phytocompounds for cancer prevention and therapy. Phytochemicals affect proliferation, differentiation, apoptosis evasion, and angiogenesis of cancer cells. Such phytoconstituents are beneficial by reducing the harmful impacts of chemotherapy and increasing the viability of traditional chemotherapy [21]. Dietary phytochemicals prevent carcinogenesis, a multistage mechanism that involves tumor formation, growth, malignant development, and progression. The mechanism of action of many phytochemicals that suppress the development of tumors in animal models involve modulation of molecular signal transduction pathways that react to various external signals, and are implicated in cancer development. 

A significant signaling pathway implicated in initiating changes in gene expression is the Janus kinase (JAK)-signal transducer and activator of transcription (STAT) pathway. Aberrated activation of this signaling pathway promotes tumorigenesis. Abnormal activation occurs when a ligand binds to its receptor constitutively or when tyrosine kinase is inappropriately stimulated [22]. The JAK/STAT signaling may act either directly or indirectly by triggering nuclear factor-κB (NF-κB) activation [23]. Blocking JAK/STAT signaling in cancer cells can suppress the expression of target genes that control essential cell functions and hamper cancer cells from evading growth control mechanism, such as apoptosis and invasion. Therefore, antagonizing JAK/STAT signaling could obstruct the transformation of preneoplastic lesions into a malignant tumor. 

Studies have demonstrated that numerous phytochemicals can interfere with the JAK/STAT signaling mechanism in human malignant cells. There are only a few previous reviews which provide an in-depth analysis of this interesting research area. Most of the prior publications focus either on a single phytochemical [24,25,26] or natural compounds that modulate various transcription factors, including STATs [27,28,29,30,31]. In this context, this review article aims to explore various naturally-occurring phytoconstituents that can suppress the JAK/STAT signaling pathway, thereby inhibiting growth of cancer cells caused by aberrated JAK/STAT signaling. We have presented and critically analyzed up-to-date research conducted on this topic to identify possible chemopreventive and therapeutic agents with established molecular targets.

## 2. Literature Search Methodology

We have followed the preferred reporting items for systematic reviews and meta-analysis (PRISMA) guidelines to perform literature searches [32]. In vitro, in vivo, and human studies that explored the antiproliferative potency of phytoconstituents by inhibiting JAK/STAT pathway were screened using authentic databases, such as PubMed, ScienceDirect, Web of Science, SpringerLink, Scopus, and Google Scholar. Relevant full articles published until January 2020 in peer-reviewed journals have been included. Conference abstracts, books, book chapters, and unpublished findings have not been included. Only papers published in the English language have been considered and included in this review. The major keywords used for literature search included cancer, tumor, proliferation, JAK, STAT, phytochemicals, polyphenols, terpenoids, alkaloids, saponins, steroids, in vivo, in vitro, and clinical studies.

## 3. JAK/STAT Signaling Pathway

The JAK/STAT signaling pathway serves as an important agent involved in functional responses. Three main proteins are involved in JAK/STAT signaling, including cell-surface, JAK, and STAT receptors. Upon activation of the receptor complex, STAT is phosphorylated on its retained tyrosine residues, which prompts the release of STAT from the receptor and the dimerization of two STAT molecules. The dimer then translocates into the nucleus wherein the dimer gets attached to DNA and stimulates the expression of genes that are responsive to cytokines [33,34]. Cytokines, including interferon (IFN) and interleukin (IL), attach to their unique receptors involved in the JAK/STAT pathway. This binding induces JAK to phosphorylate one another and afterward phosphorylate the receptor by itself at the receptor STAT protein binding sites. After binding, STAT is phosphorylated by JAK and separated from the receptor where they then bind with another STAT molecule to form a dimer. The STAT complex then travels into the nucleus, gets attached to DNA, and initiates gene expression that promotes tumor cell proliferation and survival (Figure 1). Such cytokines bind specific cytokine receptors, which are related to the JAK lineage [35,36]. Four types of mammalian JAKs exist, and they include tyrosine kinases 2 (TYK2), JAK1, JAK2, and JAK3. 

Transcriptional activity and its functional roles are needed for dimerization of STATs through phosphorylation of tyrosine residues. This promotes and mediates binding of regulatory receptors with the formation of phosphorylated complex [37]. STAT dimerization regulates the process of phosphorylation by associating with the SH2 domain. In the 1990s, it was found that STAT proteins are IFN-directed genes [33,34,38]. Among primates, STATs consist of seven structurally distinct members: STAT1, STAT2, STAT3, STAT4, STAT5A, STAT5B, and STAT6 [39]. STAT1, STAT3, and STAT5 were elevated in various human cancer cells [40]. Such proteins involve cytoplasmic transcription factors; for example, transduction of signals from hormones, cytokines, and growth factors. Furthermore, STATs have downstream effector roles [39,41]. STAT proteins regulate numerous cellular biological functions, including organogenesis, fetal growth, programed cell death, differentiation, growth, inflammation, and the immune system [42,43,44,45,46]. Inside the cytoplasm, STAT proteins remain as monomers; however, they produce dimers after phosphorylation of their tyrosine domains [37]. The dimer travels to the nucleus to empower transcriptional functions. In the formation of tumors, STATs are constitutively stimulated by tyrosine kinases, including Cluster-Abelson breakpoint (Bcr-Abl), JAK, Src, and epidermal growth factor receptor (EGFR).

Unregulated JAK/STAT signaling contributes to proliferation, survival, inflammation, invasion, new blood vessel formation, and metastasis, which are implicated in cancer initiation, progression, and advancement [27].

## 4. JAK/STAT Inhibitors

### 4.1. JAK Inhibitors

In 1996, it was revealed that AG490, a pan-JAK inhibitor, had the ability to impede the occurrence of recurrent B-cell leukemia based on in vitro and in vivo studies [47]. Since then, natural components, including resveratrol, curcumin, piceatannol, and flavopiridol, have been preclinically examined and found to inhibit innumerable inflammatory pathways [48]. These pathways inhibit STAT3 phosphorylation, primarily through decreased cytokine production or as a direct JAKs inhibitor [49,50]. Further, after the JAK2 mutation was discovered in myeloproliferative disorders, orally bioavailable and more effective JAK inhibitors have been developed. According to preclinical studies in myeloproliferative models, these compounds are incredibly efficient in their ability to abrogate illness, which has stimulated clinical testing [51,52,53]. The well-known JAK1/2 inhibitor INCB018424 (Incyte Corp., Wilmington, DE, USA) has demonstrated huge clinical benefits (reduction of splenomegaly, night sweats, weakness, and irritation) correlating with a decline in proinflammatory cytokines release. INCB018424 is in the third phase of clinical trials [52]. Merely marginal reductions were observed in the burden of the mutated JAK2 allele. This may be due to insufficient suppression of the JAK2 kinase mutant, or because the disease is the responsibility of another signaler. The additional JAK inhibitors (XL019, CEP-701, and TG101348) have shown similar results; however, they possess different adverse events, which include anemia, thrombocytopenia, neutropenia, transaminitis, neurotoxicity, and gastrointestinal intolerance [52]. The potency, sensitivity (JAK1/2 versus JAK2), and half-life of both JAK inhibitors are the factors underlying their side effect variations. Critically, JAK1/2 is required for natural hematopoiesis and thrombocytopenia, and anemia may result from these medications unless specific dosing schedules or lower doses are provided [54]. AZD1480, a JAK 1/2 inhibitor, was evaluated in the treatment of interleukin-6 (IL-6)-driven cancers, including prostate, breast, and ovarian cancer. The results of this study showed that AZD1480 suppressed tumor growth [55]. Such drugs are presently being studied for solid tumors, including lymphomas, sarcomas, and carcinomas, in phase 1 clinical trials.

### 4.2. STAT Inhibitors

An ideal STAT3-binding site (like double-stranded DNA) was made and injected intratumorally or intravenously to cultured cells to aim STAT3 as a DNA-binding protein. It prevents the STAT3 dimer from binding to its targets by isolating it into the STAT3 decoy [56,57,58,59]. Elevation of tyrosine phosphorylated STAT3 was analyzed preclinically by the STAT3 decoy in neck and head squamous cell carcinomas. The significant levels of tyrosine-phosphorylated STAT3 resulted in cancer cell apoptosis, resulting in lower tumor growth. Interaction between the decoy and other therapies has also been demonstrated. Currently, this STAT3 decoy is clinically studied in patients with head and neck cancer, and is less intrusive and readily accessible for local injection.

Direct STAT3 inhibitors have been explored, and the research has concentrated on designing agents influencing the SH2 domain by restricting STAT3 phosphorylation and/or dimerization. The primary STAT3 inhibitors, such as peptidomimetics and engineered small molecules, act by inhibiting phosphorylation of intracellular STAT3, DNA-binding, and transcriptional activities [60]. In comparison, peptidomimetics may be associated with nonphosphorylated monomeric proteins of STAT3 via pY-SH2 domain binding (the peptide or mimetic incorporates the pY motif in the SH2 domain of the STAT3 monomer) to create a hetero complex. It decreases the rates of free nonphosphorylated STAT3 monomers usable for phosphorylation and de novo activation. Some of these agents have decreased the progression of cancer in various preclinical models [60,61,62]. Regardless of the fact that those components demonstrated a fair sensitivity to repress the activity of STAT3, they were not clinically tested because of the high concentrations required for them to be effective [63]. The utilization of a chemical database of previously synthesized compounds in a luciferase reporter cell-based analysis prompted the discovery of transcription-dependent STAT3 and STAT5 inhibitors [64,65,66]. For example, nifuroxazide, a drug used to treat diarrhea, could effectively inhibit JAK2 and TYK2, while decreasing the p-STAT3 levels in multiple myeloma [67]. The antimalarial drug, pyrimethamine, has been described as an STAT3 inhibitor and a myeloma growth inhibitor. It is currently undergoing clinical trials to treat lymphocytic leukemia [65]. 

The appearance of mutant (dominant-negative forms) STATs has allowed for delineation of the specific function of domains or residues, and it has led to the discovery of the novel or noncanonical functions for the STATs as mediators of tumorigenesis [68,69]. It has been suggested that both noncanonical and canonical functions of individual STAT play a central role in the initiation and promotion of tumors. The canonical pathway is characterized as tyrosine phosphorylated STATs functioning as transcription factors. The noncanonical signaling embodies various functions of nontyrosine phosphorylated STATs, including mediation of DNA methylation, focal adhesions activation, and regulation of mitochondrial functions [70,71,72,73]. The noncanonical mechanism represents the activation of nontyrosine phosphorylated STAT3, inducing transcription in combination with NF-κB or CD44, which is situated in the mitochondria to control the synthesis of ATP, and then interacts with the stathmin (a microtubule-linking protein) to inflect cell motility [74,75]. STAT3–tubulin associations may be affected by microtubule-targeting agents, e.g., paclitaxel [76]. Moreover, STAT3 phosphorylated on its serine residue, but not on its tyrosine residue, can control transcription in chronic lymphoblastic leukemia [77]. STAT acetylation, ubiquitylation, and sumoylation functions are being examined in controlling tumor growth and metastasis. Subsequently, both nontyrosine and tyrosine phosphorylated STAT3 perform significant roles in cancer cells, and this knowledge can be utilized to develop potential anticancer agents.

## 5. Phytochemicals Inhibiting the JAK/STAT Signaling Pathway

Several bioactive phytocompounds have demonstrated that the JAK/STAT pathway is inhibited by various mechanisms. The JAK/STAT pathway comprises more than one site of action that phytochemicals can target. Phytochemicals can block this signaling pathway by reducing the levels of cytokines or growth hormones which trigger JAK/STAT protein activation. Phytochemicals can also act by stopping JAK phosphorylation before STAT activation. Controlling JAK/STAT signaling can be achieved by inhibiting STAT dimerization and by preventing the translocation of STAT dimer from the cytoplasm into the nucleus. The ultimate goal in the signaling pathway is to hinder the STAT–DNA binding, which directly inhibits JAK/STAT-regulated gene transcription. Phytochemicals may also control the JAK/STAT pathway by association with JAK/STAT pathway inhibitors, such as SHP [78,79,80,81]. Few phytochemicals can specifically block one target site, while others can control the JAK/STAT system at multiple sites. In the following sections, we analyze the experimental evidence of the capacity of various phytochemicals from diverse plant origin to regulate JAK/STAT pathway in various preclinical cancer models.

### 5.1. Phenolics and Polyphenols

These phytocompounds comprise 15-C atoms as well as one or several hydroxyl groups attached to one or multiple aromatic rings and they are water soluble. More than 8000 phenolic components exist in nature, with many found in various vegetables and fruits. These are metabolites of plants that are considered to confer numerous health benefits via different biochemical and pharmacological actions, such as anti-inflammatory, immunomodulatory, antioxidant, and antimicrobial actions, as well as regulation of numerous cell signaling pathways.

#### 5.1.1. Resveratrol 

Resveratrol (also known as, *trans*-3,5,4′-trihydroxystilbene, Figure 2), a compound present in peanuts, berries, and grapes, has been demonstrated as a potent chemopreventive and chemotherapeutic agent [82,83,84,85]. Resveratrol’s antiproliferative and cytotoxic functions have been associated with JAK/STAT pathway inhibition. Resveratrol obstructed JAK phosphorylation and, as a result, blocked STAT1 phosphorylation in human epidermoid carcinoma (A431) cells [86] (Table 1). The growth of human multiple myeloma cells was inhibited by resveratrol to overcome the chemoresistance by suppressing both the inducible and constitutive activation of STAT3, resulting in downregulation of antiapoptotic gene expressions [87]. Besides the inhibition of the JAK/STAT signaling pathway, resveratrol was also found to suppress the function of Src tyrosine kinase, which consequently blocked STAT3’s action in specific cancer cells [88]. Resveratrol was effective in blocking both signaling channels for JAK/STAT and Src/STAT in in vitro experiments. A resveratrol analog, 3,4,5,4′-tetramethoxystilbene, repressed STAT3 phosphorylation and exhibited greater antitumor activity than resveratrol [89]. An experimental study showed that resveratrol was bound to the estrogen receptor, but it did not cause the proliferation of estrogenic cells. Interaction with resveratrol has been found to result in a viable IL-6 transporter, a potent STAT3 mediator [90]. It is important to note that resveratrol studies have been concentrated on the short-term results with constrained clinical research.

#### 5.1.2. Curcumin

Curcumin is one of the most studied polyphenols for anticancer potential [91,92,93,94]. Curcumin is the key curcuminoid present in *Curcuma longa* L., (family Zingiberaceae). Curcumin is known to regulate the JAK/STAT signaling pathway. Unlike resveratrol, curcumin blocked STAT3 phosphorylation, thereby blocking STAT3 dimer translocation from cytoplasm into the nucleus of human multiple myeloma cells [95]. Curcumin has been found to suppress STAT1, STAT3, JAK1, and JAK2 phosphorylation in microglia stimulated with gangliosides, lipopolysaccharides (LPS), and/or microglia cells. It was also found that activation of Src homology region 2 domain-containing phosphatase-2 (SHP-2), a negative JAK behavior regulator, was the probable mechanism influencing the JAK/STAT signaling pathway for curcumin-mediated inhibition [78]. Additional research showed curcumin’s potential to reduce STAT3 phosphorylation in small-cell lung cancer cells by desensitizing the downstream target genes, which was accountable for cancer cell proliferation [96]. The constitutive activation of the JAK/STAT pathway was also disrupted by curcumin in T-cell leukemia, in which it decreased JAK and STAT phosphorylation and resulted in growth arrest and subsequent apoptosis [97,98].

#### 5.1.3. Ascochlorin

Ascochlorin, an isoprenoid antibiotic obtained from the fungus *Ascochyta viciae* Lib., has exhibited significant antiproliferative response in many tumor cell lines and in vivo experimental models [99]. Ascochlorin has been found to suppress the migration and invasion in two human glioblastoma cell lines (A172 and U373MG cells) by decreasing matrix metalloproteinase-2 gelatinolytic action and expression. To understand the underlying mechanism by which the ascochlorin smothered cell invasion and migration, the effect of ascochlorin on JAK/STAT signaling has been explored. Ascochlorin essentially diminished JAK2/STAT3 phosphorylation and blocked translocation of STAT3 to the nucleus [100].

#### 5.1.4. Bergamottin

Bergamottin, a furanocoumarin, is present in grape juice, lime, lemon, and bergamot oils. It is a potential antioxidant, anti-inflammatory, and anticancer agent [101]. In human multiple myeloma (U266) cells, bergamottin has been found to suppress a constitutive activation of STAT3. This effect was achieved through resisting phosphorylation of JAK 1/2 and c-Src. Furthermore, bergamottin induced the expression of the tyrosine phosphatase SHP-1 and the silencing of the SHP-1 gene by siRNA abrogated the capability of bergamottin to hinder STAT3 actuation, which is a crucial factor in bergamottin’s action on SHP-1. It has also been associated with downregulation of the expression of STAT3-directed genes, such as VEGF, COX-2, survivin, cyclin D1, Bcl-xL, Bcl-2, and inhibitor of apoptosis protein-1 (IAP-1). This is related to substantial increase in apoptosis associated with cell cycle arrest at sub-G1 phase and poly (ADP-ribose) polymerase (PARP) cleavage induced by caspase-3 [102].

#### 5.1.5. Capillarisin

One of the principal bioactive components of *Artemisia capillaries* Thunb., capillarisin is a naturally-occurring chromone. It is an analogue of benzopyran, and a coumarin isomer. It explicitly restrained both constitutive and inducible STAT3 activation at tyrosine 705, but not at serine 727, in human multiple myeloma cells [103]. Other than hindering phosphorylation of STAT3, capillarisin hindered constitutive STAT3 movement and nuclear translocation. The concealment of STAT3 was interceded through the restraint of the activation of upstream JAK1, JAK2, and c-Src kinases. Interestingly, knockdown of the SHP-1 and SHP-2 genes by siRNA depressed the capability of capillarisin to hinder the activation of JAK1 and STAT3. This suggests that SHP-1 and SHP-2 play a vital role in its potential mechanism of action. Apart from this, capillarisin also downregulated the STAT3-controlled antiapoptotic (Bcl-xL, Bcl-2, survivin, and IAP-1) and proliferative (cyclin D1) gene expression levels, resulting in decrease in cell viability levels, cell cycle arrest at sub-G1 phase, and induction of cell apoptosis. Overall, capillarisin represents a potent inhibitor of STAT3 that can have a negative regulatory effect on the chemoresistance and metastasis of neoplastic cells [103].

#### 5.1.6. Bavachin

Bavachin, a flavonoid, is derived from the seeds of *Psoralea corylifolia* Linn., (currently known as *Cullen corylifolium* (L.) Medik.). It is a phytoestrogen, which activates both the estrogen receptors, namely estrogen receptor-α (ERα) and ERβ. Bavachin diminished the proliferation of multiple myeloma cells; however, it is not cytotoxic to normal cells. It restrained the activation of STAT3 in multiple myeloma cell lines. Moreover, bavachin expanded the gene expression of phorbol-12-myristate-13-acetate-induced protein 1 (PMAIP1, alternative known as Noxa) and p53. It also diminished the expression of Bcl-2, X-linked inhibitor of apoptosis protein (XIAP), survivin, and B-cell lymphoma-extra-large (Bcl-xL). Furthermore, bavachin was found to induce apoptosis through caspase-3 and caspase-9 activation, confirming the participation of mitochondrial pathway. Overall, bavachin targets STAT3 and could be used for the treatment of multiple myeloma [104].

#### 5.1.7. Epigallocatechin Gallate (EGCG)

EGCG is the catechin commonly present in tea (*Camellia sinensis* (L.) Kuntze), and it has demonstrated promising anticancer properties against many types of tumors [105,106,107,108]. A study conducted by Tang et al. (2012) [109] found that both the JAK3 and STAT3 protein phosphorylation and translation were diminished by EGCG in two human pancreatic cancer cell lines, namely AsPC-1 and PANC-1. The expressions of STAT3-directed genes in pancreatic cancer cells were also inhibited by EGCG. By subduing the bioavailability of insulin-like growth factors (IGFs), it also decreased the level of IGF-mediated phosphorylated-STAT3 proteins in hepatocellular carcinoma cells [110]. Several investigators have found that EGCG treatment suppressed the action of STAT3 in breast, head, and neck carcinoma cells [111,112]. Very recently, EGCG has been found to inhibit the proliferation of chronic myeloid leukemia cells by inducing apoptosis via inhibition of the Bcr/Abl oncoprotein and controlling its downstream pathways, such as p38-MAPK/JNK and JAK2/STAT3/AKT [113].

#### 5.1.8. Emodin

Emodin (1,3,8-trihydroxy-6-methyl anthraquinone) is an anthraquinone obtained from Japanese knotweed, buckthorn, and rhubarb. It is also explicitly extracted from *Rheum Palmatum* L., as well as *Socotrine aloe*, *Barbados aloe*, and *Zanzibar aloe*. Additionally, it is found in numerous fungi species, including members of genera Pestalotiopsis, Pyrenochaeta, and Aspergillus. Emodin is known to possess antioxidant, anti-inflammatory, immunomodulatory, and antitumor properties [114,115,116]. A study was conducted by He et al., [117] with 26S proteasome inhibitors by utilizing cell-based screening test in which emodin was reported as an effective human 26S proteasome inhibitor. Emodin hindered chymotrypsin-like and caspase-like actions of the human 26S proteasome and enhanced the ubiquitination of endogenous cell proteins. Molecular docking studies demonstrated that the orientation/conformation in the active pocket of the 1, 2, and 5 subunits of 26S proteasome was suitable for nucleophilic attack. Emodin stimulated the antiproliferative role of interferon α/β (IFN-α/β) by increasing the STAT1 phosphorylation, diminishing the STAT3 phosphorylation, and expanding the expression of an endogenous gene activated by IFN-α. It also repressed IFN-α-stimulated ubiquitination and type I interferon receptor 1 (IFNAR1) degradation. Moreover, it promoted the antiproliferative impact of IFN-α against HeLa cells (a cervical carcinoma cell line) and diminished the development of tumors in Huh7 hepatocellular carcinoma-bearing mice. Based on the results, it is possible that emodin enhanced the antiproliferative action of IFN-α/β by the inhibition of the JAK/STAT signaling pathway through restraining 26S proteasome-stimulated IFNAR1 degradation [117].

#### 5.1.9. Chalcones

Chalcones are flavonoids found in various fruits (oranges, strawberries, and tomatoes), vegetables (bean sprouts, shallots, and potatoes), and spices (licorice). Chalcones have been shown to be promising chemopreventive and antitumor agents due to their antioxidant, cytotoxic, apoptosis-inducing, and numerous cell signaling-modulatory properties [118,119]. Chalcone, a α,β-unsaturated flavonoid, restrained the phosphorylation of STAT3 in LPS- and IL-6-regulated endothelial cells [120].

#### 5.1.10. Formononetin

Formononetin, a naturally-occurring isoflavone, may be found in low quantities in numerous dietary products, such as beans, carrot, cauliflower, iceberg lettuce, green peas, and red potatoes. It is associated with the Fabaceae family and is isolated from the roots of *Astragalus mongholicus* Bunge. This root represents an essential ingredient utilized in the traditional Chinese medicine due to its suppressive effect against various malignant tumors. Formononetin has been under intense investigation during the last decade due to its ability to promote apoptosis and suppress proliferation in multiple in vitro and in vivo experimental cancer models, such as breast, colorectal, and prostate carcinoma [121,122]. One investigation demonstrated that formononetin effectively repressed the proliferation and invasion of HCT116 and SW1116 colon carcinoma cells. It also caused cell cycle arrest at the G0–G1 stage through the downregulation of protein expressions of cyclins D1. The anticancer impact of formononetin was found to be mediated by the impairment of phosphoinositide 3-kinase (PI3K)/protein kinase B (Akt) and STAT3 signaling pathways [123].

#### 5.1.11. Garcinol

Garcinol, a poly-isoprenylated benzophenone analog, is isolated from the dried fruit rind of *Garcinia indica* Choisy. The antioxidant, anti-inflammatory, and anticancer properties of garcinol underscore its therapeutic benefits [124,125]. It may stop constitutive and IL-6-inducible STAT3 activation in hepatocellular carcinoma cells. The molecular docking analysis demonstrated garcinol’s capability to link with STAT3 at the SH2 domain and to depress its dimerization. It additionally repressed the acetylation of STAT3 and subsequently disabled its DNA binding capacity as a result of an acetyltransferase inhibitor. The restraint of STAT3 actuation by garcinol prompted repression of the expression levels of different genes associated with proliferation, survival, and angiogenesis. Moreover, it repressed the proliferation of hepatocellular carcinoma cells by stimulating apoptosis. Astoundingly, garcinol hindered the development of human hepatocellular carcinoma xenograft tumors in athymic nu/nu mice by impairing STAT3 activation. The proapoptotic and antiproliferative impacts of garcinol in hepatocellular carcinoma were mediated through inhibition of the STAT3 signaling pathway [126].

#### 5.1.12. Cardamonin

Cardamonin (chemically known as 3,4,2,4-tetrahydroxychalcone) is a chalconoid that has been found in a few plants, such as *Alpinia conchigera* Griff. and *Alpinia hainanensis* K.Schum. It has received considerable attention from researchers because of its putative health benefits, including anticancer potential [127]. A study investigated the impact of cardamonin on glioblastoma stem cells, and examined its effect on apoptosis and self-renewal and whether its activity is related to the STAT3 pathway in this cancer cell population. CD133-positive (CD133+) glioma stem-like cells (GSCs), a glioblastoma stem cell line was generated from human glioblastoma tissues. In CD133+ GSCs, cardamonin hindered proliferation and promoted apoptosis. The proapoptotic impacts of temozolomide were additionally elevated in vitro via cardamonin in U87 cells and CD133+ GSCs. Cardamonin additionally repressed STAT3 initiation by luciferase assay and stifled the downstream STAT3 gene expressions, for example, Bcl-xL, Mcl-1, Bcl-2, survivin, and VEGF. Besides, cardamonin halted STAT3’s travel to the nucleus and dimerization in CD133+ GSCs. Apart from this, molecular docking study affirmed that cardamonin, with a favorable binding energy of −10.78 kcal/mol, was bound to the active sites of STAT3 [128]. All these results demonstrate that cardamonin has the ability to be a new anticancer agent for glioblastoma by virtue of its ability to act as a STAT3 inhibitor. 

#### 5.1.13. Caffeic Acid

Caffeic acid and phenethyl ester caffeic acid (CAPE) are phenolic compounds synthesized by different plant organisms. They are present in beverages, such as wine, tea (*Camellia sinensis* (L.) Kuntze) and coffee (*Coffea arabica* L.). Caffeic acid and CAPE are known to have anti-inflammatory, antioxidant, and anticancer properties [129,130]. Caffeic acid blocked STAT3 phosphorylation by suppressing Src tyrosine kinase, which inhibited the proliferation and caused the downregulation of VEGF in the human renal carcinoma cells [131].

#### 5.1.14. Silibinin

Silibinin, a flavanone present in Silybum marianum (L.) Gaertn. (milk thistle), is an extraordinary hepatoprotective agent with anticancer potential [132,133,134]. Silibinin’s benefits were also observed when used in conjunction with traditional chemotherapy since they decreased adverse effects of anticancer drugs (e.g., nephrotoxicity, neurotoxicity, and cardiotoxicity) while preventing or even reversing chemotherapy resistance [24]. Silibinin blocked STAT3–DNA attachment, inhibited constitutively active STAT3 phosphorylation in prostate cancer cells, and induced caspase activation that contributed to cell death [135]. This research showed that in conjunction with silibinin, piceatannol (a JAK1 inhibitor) lessened STAT3 phosphorylation at Tyr705 and thus triggered caps, which induced the programmed cell death of prostate cancer cells [135]. Silibinin has also been shown to suppress STAT3 phosphorylation and thus decrease its transcriptional function in urethane-induced lung tumors in A/J mice [136]. 

#### 5.1.15. Butein

A flavonoid named butein is isolated from the bark of *Toxicodendron vernicifluum* (Stokes) F.A. Barkley (formerly *Rhus verniciflua* Stokes.) and the flowers of *Butea monosperma* (Lam.) Kuntze. Butein, a multitargeted agent, is considered to have antioxidant, anti-inflammatory, hypotensive, antidiabetic, anticancer, and neuroprotective effects [137,138]. In multiple myeloma cells, it restrained the constitutive and IL-6-induced activation of STAT3 through inactivation of JAK1 and c-Src in a concentration-dependent fashion [139]. Butein also suppressed the growth of xenografted human hepatocellular carcinoma in male nude mice [140].

#### 5.1.16. 5,7-Dihydroxyflavone

5,7-Dihydroxyflavone is a natural flavonoid present in propolis and honey, in addition to various herbs, such as *Passiflora incarnate* L., *Passiflora caerulea* L., and *Oroxylum indicum* (L.) Kurz. Various in vivo and in vitro investigations have reported its potential for chemoprevention and therapy of various cancers [141,142]. By reducing Akt and STAT3 phosphorylation, 5,7-dihydroxyflavone significantly suppressed the growth of human hepatocellular carcinoma (HepG2) tumor xenografts [143].

#### 5.1.17. Honokiol

Honokiol, a phytochemical present in plants such as *Magnolia officinalis* Rehder & E.H.Wilson, and *Magnolia grandiflora* L., exhibited antiproliferative, proapoptotic, and cell-cycle-modulatory activities in gastric, colon, colorectal, hepatocellular, pancreatic, lung, glioblastoma, melanoma, leukemia, skin, breast, ovarian, renal, prostate, head and neck, mesangial, and squamous cell carcinoma through modulation of multiple oncogenic targets [144,145,146,147,148]. Honokiol diminished the generation of p-STAT3 tyr705 and p-JAK2 Tyr1007 in human monocytic cell line (THP1 cells), and human erythroleukemia cell line (HEL cells) [149]. It additionally restrained both constitutive and inducible STAT3 activation, resulting in other downstream impacts, such as the increased expression level of SHP1. However, SOCS1/3 protein levels and knockdown of SHP1 reversed honokiol-induced STAT3 signal hindrance [149]. In addition, honokiol decreased the percentage of mRNA of STAT3 target genes (cyclin D1, Bcl-2, surviving, and Bcl-xL) in a concentration-dependent fashion in myeloid leukemia cells [149].

#### 5.1.18. Casticin

Casticin can be found in several plants, such as Achillea millefolium L., Artemisia abrotanum L., Camellia sinensis (L.) Kuntze, Centipeda minima (L.) A.Braun & Asch., Clausena excavate Burm.f., Crataegus pinnatifida Bunge., Croton betulaster Müll Arg., Daphne genkwa Siebold & Zucc., Ficus microcarpa L.f., Nelsonia canescens (Lam.) Spreng., Pavetta crassipes K.Schum., Vitex trifolia subsp. Litoralis steenis (formerly Vitex rotundifolia L.), Vitex agnus-castus L., Vitex negundo L., and Vitex trifolia L. Recent experimental evidence underscores the ability of casticin as a candidate for antineoplastic drugs [150,151]. It showed antiproliferative, cell-cycle-regulatory, and proapoptotic activities in breast, cervical, colon, colorectal, gastric, liver, leukemia, lung, pancreatic, ovarian, and prostate cancer cells [152,153]. It also promoted apoptosis, inhibited constitutively active STAT3, modulated STAT3 activation by modifying the activity of upstream STAT3 regulators, and abrogated IL-6-induced STAT3 activation, thereby suppressing the JAK/STAT pathway in various cancer cells, such as tongue squamous cell carcinoma (HN-9), hypertriploid renal cell carcinoma (786-O), and oral squamous cell carcinoma (YD-8) [154].

#### 5.1.19. Apigenin

Found plentifully in many medicinal plants and in many vegetables and fruits, apigenin, a naturally-occurring flavonoid, has several physiological functions [155,156]. It showed anticancer activity via JAK/STAT pathway inhibition by reducing phosphorylated JAK1/2 and STAT3 in breast cancer (BT-474) cells [25]. Apigenin induced programmed cell death of colon cancer cells by hindering STAT3 phosphorylation and hence downregulated the antiapoptotic proteins Mcl-1 and Bcl-xL [157].

#### 5.1.20. Wedelolactone

Wedelolactone is a natural coumarin with promising anticancer activities [158]. Wedelolactone, obtained from *Eclipta prostrata* (L.) L. (formerly *Eclipta alba* (L.) Hassk.), and *Sphagneticola calendulacea* (L.) Pruski (formerly *Wedelia calendulacea* (L.) Pruski)*,* exhibited antiproliferative and apoptosis-inducing effects in breast, prostate, neuroblastoma, pancreatic, mammary carcinosarcoma, myeloma, and leukemia cells [159,160]. This prevented the dephosphorylation of STAT1 and prolonged the activation of STAT1 by mutual inhibition of T-cell protein tyrosine phosphatase (TCPTP) in p-GAS-HepG2 cells [161].

### 5.2. Terpenoids

The terpenoids, also known as isoprenoids, are one broad family of naturally occurring, terpene-derived organic compounds. With more than 40,000 structurally diverse components, the terpenoids are the largest group of secondary metabolites from the plant kingdom. The majority of terpenoids with structural differences are biologically active and are used to treat numerous diseases around the world. Several terpenoids have been developed as anticancer drugs, including taxol. Natural terpenoids and related compounds have inhibited the proliferation of various human carcinoma cells and suppressed the tumor development in experimental animals [162,163,164]. Several phytoconstituents belonging to this category can inhibit the JAK/STAT pathway by inhibiting the phosphorylation of STAT3. Various terpenoids even inhibited the phosphorylation of JAK1, JAK2, and c-Src in various carcinoma cell lines. The role of various terpenoids in the JAK/STAT signaling pathway in various cancers is the focus of the subsequent sections.

#### 5.2.1. Cucurbitacins

Cucurbitacins are associated with tetracyclic triterpenoids found in the Cucurbitaceae family. There are 17 major molecules in the cucurbitacins group, which range from cucurbitacin A to cucurbitacin T with hundreds of analogs [165]. Cucurbitacins and their derivatives have various biological and pharmacological actions, including anticancer effects [166,167]. Cucurbitacin B (Figure 3) demonstrated anti-STAT3 effects in human pancreatic cancer cells [168,169] and leukemia cells [170] (Table 2). Even though cucurbitacin E decreased the amount of phosphorylated STAT3 in human bladder cancer cells, it did not alter the levels of STAT3 [171]. Cucurbitacin I (JSI-124) reduced the phosphorylation of STAT3 and JAK in Src-transformed fibroblast and in human lung carcinoma cells. This subsequently impaired STAT3–DNA binding [172]. Likewise, the phosphorylated levels of STAT3 in glioblastoma multiforme cells treated with cucurbitacin I were greatly decreased. This resulted in G2/M cell cycle suspension by downregulation of the expression of cdc2 and cyclin B1 [173]. Moreover, the inhibitory impact on JAK/STAT by cucurbitacin I was additionally determined in Sézary (Sz) syndrome [174] and anaplastic large cell lymphoma [175]. Cucurbitacin Q hindered STAT3 levels; however, it did not influence the activation of JAK [176]. Based on the aforementioned reports, various cucurbitacins exhibit distinct mechanisms to function as JAK and STAT inhibitors. 

#### 5.2.2. Andrographolide

Andrographolide, a diterpenoid lactone extracted from *Andrographis paniculata* (Burm.f.) Nees, has strong anticancer and cancer preventive potential [177,178]. A study performed by Zhou et al. [179] analyzed the impact of andrographolide on the STAT3 pathway and assessed whether restraint of STAT3 action by andrographolide could sensitize MDA-MB-231 breast cancer cells to doxorubicin, a chemotherapeutic drug. Andrographolide can stifle both constitutively activated and IL-6-induced phosphorylation of STAT3 and, consequently, its nuclear translocation into cancer cells. Such restraint was accomplished through restraint of JAK1/2 and interaction between STAT3 and gp130. To comprehend the importance of the inhibiting effect of andrographolide on STAT3, the impact of it on doxorubicin-induced apoptosis via promoting caspase activation and apoptotic pathway has been evaluated in human cancer cells (HeLa, HepG2, HCT116, and MDA-MB-231). In that test, the constitutive activation level of STAT3 was found to correspond to the interference of malignant cell growth in doxorubicin-induced apoptosis. The long-term colony formation assay in addition to the short-term cell proliferation assay demonstrated that andrographolide significantly promoted doxorubicin-induced cell death in cancer cells, showing that andrographolide improves the sensitivity of cancer cells to doxorubicin principally by means of STAT3 suppression. These results uncover a novel anticancer function of andrographolide in cancer treatment.

#### 5.2.3. Betulinic Acid 

A natural pentacyclic triterpenoid, betulinic acid is extracted from the exterior bark of white birch plants. In recent years, betulinic acid has gained considerable attention due to its strong cytotoxic activity against a variety of tumor cells [180,181]. Betulinic acid hindered constitutive activation of STAT3, Src kinase, and JAK1/2 [182]. Betulinic-acid-induced expression of the PTP SHP-1 and silencing of the SHP-1 gene nullified the capability of betulinic acid to hinder STAT3 initiation and salvages betulinic-acid-induced cell death. It also downregulated the expression level of STAT3-directed genes, such as survivin, Bcl-2, Bcl-xL, and cyclin D1. There was also a rise in apoptosis as evidenced by cell cycle arrest at sub-G1 phase and increase in PARP cleavage induced by caspase-3. In addition, overactivation of constitutive dynamic STAT3 fundamentally diminished the betulinic acid-induced apoptosis [182].

#### 5.2.4. γ-Tocotrienol

γ-Tocotrienol is one of the four types of tocotrienol, a type of vitamin E. γ-Tocotrienol has established potency in preclinical anticancer research [183,184]. There are eight forms of vitamin E, each consisting of a head segment associated with either a saturated (phytyl) or unsaturated (farnesyl) tail. γ-Tocotrienol prevented the phosphorylation of molecules involved in the JAK/STAT pathway, including JAK1, JAK2, and c-Src, in human liver cancer cells [185]. It also induced the expression of SHP-1, which is one type of tyrosine phosphatase engaged in the JAK/STAT signal’s negative regulation [79], in liver cancer cells in a concentration-dependent fashion [185,186].

#### 5.2.5. Cryptotanshinone

Cryptotanshinone, a quinoid diterpene, has been isolated from *Salvia miltiorrhiza* Bunge. Experimental investigations have demonstrated evidence that cryptotanshinone could be used as an anticancer agent [187]. In pancreatic carcinoma cells, cryptotanshinone repressed STAT3 phosphorylation via a mechanism autonomous of JAK2 phosphorylation [188]. It was found to bind directly with STAT3 and prevented STAT3’s dimerization. Computational modelling also demonstrated that cryptotanshinone has the capacity to bind with STAT3’s SH2 domain [188].

#### 5.2.6. Nimbolide

Nimbolide, a triterpene isolated from the flowers and leaves of neem (*Azadirachta indica* A. Juss.), is generally utilized in conventional medicinal practices. Emerging evidence showed that nimbolide possessed significant cytotoxic activities against various types of cancer cells and exhibited chemopreventive potential in several animal tumor models [189,190]. It can cause cell cycle arrest, most conspicuously at the G1/S stage of glioblastoma multiforme cancer cells that exhibit EGFRvIII [191]. EGFRvIII is an oncogene involved in 20% to 25% of cases of glioblastoma multiforme. Nimbolide restrained CDK4/CDK6 kinase action, which led to a decreased phosphorylation of the retinoblastoma protein. This prompted the arrest of the cell cycle at the G1/S phase and cell death [191]. Based on another investigation, nimbolide hindered STAT3 signaling and suppressed tumor formation in prostate cancer in vitro [192].

#### 5.2.7. Celastrol

A pentagonal cyclic triterpenoid named celastrol is extracted from the plant *Tripterygium wilfordii* Hook.f. It exhibits various pharmacological actions, and has antiproliferative, antiangiogenic, anti-inflammatory, proapoptotic, and antimetastatic properties [193,194]. It hindered the growth of multiple myeloma cells that are either sensitive or resistant to various chemotherapeutic drugs [195]. It also improved the proapoptotic actions of both bortezomib and thalidomide. In another study, celastrol inhibited STAT3 phosphorylation and STAT3-mediated IL-17 expression, and T-helper 17 (Th17) differentiation and proliferation in multiple myeloma cells [186].

#### 5.2.8. Ursolic Acid

A pentacyclic triterpenoid compound, ursolic acid (3β-hydroxy-urs-12-en-28-oic-acid), is present in many plants (e.g., *Mirabilis jalapa* L.) including fruits and some herbs (e.g., apples, prunes, bilberries, cranberries, basil, lavender, rosemary, peppermint, thyme, elder flower, oregano, and hawthorn). Ursolic acid has shown significant promise in the treatment and prevention of various cancers [196,197]. By suppressing Src and JAK2 phosphorylation, ursolic acid prevented the activation of STAT3 in prostate carcinoma cells [198]. Additionally, ursolic acid decreased xenografted prostate tumor development in transgenic adenocarcinoma of the mouse prostate (TRAMP) model without impacting the body weight via hindering the activation of STAT3 [198].

#### 5.2.9. Brusatol

Brusatol, a natural triterpene lactone called quassinoid, is extracted from the air-borne parts of the *Brucea javanica* (L.) Merr., plant. It has been distinguished as a strong blocker of STAT3 in preventing the progression of neck and head squamous cell carcinoma cells. Brusatol also exhibited cytotoxicity, hindered the activation of STAT3 as well as upstream kinases, including JAK1, JAK2, and Src. It diminished nuclear STAT3 and its DNA binding capability. Therefore, brusatol-mediated inhibition of STAT3 could be valuable in treating neck and head squamous cell carcinoma [199].

#### 5.2.10. Oridonin

Oridonin is a phytochemical present in *Isodon rubescens* (Hemsl.) H.Hara. Oridonin and its analogs have shown the ability to treat inflammation, neurodegeneration, and cancer [200,201]. Oridonin inhibited the proliferation of astrocytoma, breast, colorectal, hepatoma, leukemia, lung, multiple myeloma, ovarian, pancreatic, and prostate cancer cells [202,203,204,205]. It also decreased IL-6, STAT3, and p-STAT3 expression levels, implying that oridonin can downregulate the STAT3 pathway, which is accountable for the growth, invasion, and metastasis of human pancreatic carcinoma cells [206].

#### 5.2.11. Thymoquinone

Thymoquinone is the active component of black cumin (Nigella sativa L.) seed oil, and preclinical studies have established its cancer prevention effects [26,207]. Thymoquinone inhibited STAT3 phosphorylation in multiple myeloma cells, U266 and RPMI 8226 [208,209]. Thymoquinone caused downregulation of STAT3 activation in addition to a decrease in activity of JAK2 and c-Src in human gastric cancer cells. It also downregulated the expression of STAT3-controlled genes, namely survivin, cyclin D, Bcl-2, and the vascular endothelial growth factor (VEGF), and activated caspase-3, caspase-7, and caspase-9 [208].

#### 5.2.12. Parthenolide

Parthenolide, a sesquiterpene, is obtained from natural sources, such as *Tanacetum parthenium* (L.) Sch. Bip., *Tanacetum vulgare* L., *Tanacetum larvatum* (Pant.) Hayek., *Centaurea ainetensis* Boiss., and *Helianthus annuus* L. Parthenolide, a potent anticancer and anti-inflammatory agent, is presently being assessed in clinical research for cancer treatment [210]. It hindered the proliferation of bladder, pancreatic, lung, breast, skin, melanoma, glioma, skin, liver, and gastric cancer cells [211,212,213]. By suppressing JAK2 kinase activity, parthenolide also suppressed STAT3 phosphorylation induced by IL-6 in breast cancer (MDA-MB-231) cells [214].

#### 5.2.13. Dihydroartemisinin

Dihydroartemisinin, is an analogue of artemisinin extracted from *Artemisia annua* L. Artemisinin and several of its analogs, including dihydroartemisinin, have been found to induce tumor cell death via inhibiting numerous tumor-related signal transduction pathways [215,216]. Dihydroartemisinin has been found to have potent cytotoxic and proapoptotic activity by suppressing activated JAK2/STAT3 signals. It also suppressed downstream targeted proteins in the human tongue (Cal-27), colon (HCT-116), liver (HepG2) as well as head and neck (FaDu) carcinoma cell lines [217,218].

#### 5.2.14. Alantolacton

Alantolactone, a sesquiterpene, is obtained from several herbs, including *Aucklandia costus* Falc. (formerly *Aucklandia lappa* Decne.), *Inula helenium* L., *Inula japonica* Thunb. and *Inula racemose* Hook.f. It exhibits various pharmacological properties, including anti-inflammatory, antimicrobial, and anticancer activities, without significant toxicity [219]. It hindered the proliferation of glioblastoma, leukemia, prostate, colon, liver, and pulmonary carcinoma cells [220,221]. The activation of STAT3 at tyrosine 705 was greatly decreased by alantolactone in MDA-MB-231 triple-negative breast carcinoma cells. Alantolactone prevented STAT3 from traveling into the nucleus and thus affecting transcription of DNA [222].

#### 5.2.15. β-Caryophyllene oxide

β-Caryophyllene oxide is a natural bicyclic sesquiterpene. It is present in numerous essential oils, including clove oil, the oil of *Syzygium aromaticum* (L.) Merr. & L. M. Perry (cloves) stems and flowers, as well as the essential oils of *Cannabis sativa* L. (Cannabis), *Psidium guajava* L. (guava), *Origanum vulgare* L. (oregano), *Salvia Rosmarinus* Spenn. (rosemary), and *Humulus lupulus* L. The potential therapeutic utility β-caryophyllene oxide is focused mainly on its analgesic and anticancer activities [223]. A study demonstrated that the anticancer effect of β-caryophyllene oxide was mediated through interfering with the activation of STAT3 in carcinoma cells, such as multiple myeloma cells (U266) and human prostate cancer cells (DU145). The impact of β-caryophyllene oxide on STAT3 activation, related protein kinases and phosphatases, STAT3-directed gene products, and apoptosis was investigated utilizing both functional proteomics and different tumor cell lines. It was found that β-caryophyllene oxide decreased the constitutive STAT3 activation in multiple myeloma, prostate, and breast cancer cells, and has a critical concentration- and time-dependent response in multiple myeloma cells. The restraint was mediated by the inhibition of upstream kinase c-Src and JAK1/2 activation. β-Caryophyllene oxide brought about the expression of SHP-1, which is connected with the downregulation of constitutive STAT3 activation. The STAT3 activation by β-caryophyllene oxide hindered proliferation, increased apoptosis, and blunted the invasive capability of DU145 prostate carcinoma cells [224].

### 5.3. Alkaloids

Alkaloids are natural organic compounds that possess numerous pharmacological actions, including chemopreventive and chemotherapeutic effects [225,226]. These components exist in diverse plants, and several alkaloids inhibit the JAK/STAT pathway, as described below.

#### 5.3.1. Capsaicin

Capsaicin (Figure 4), an active element of chili peppers, exhibited strong anticancer properties in diverse cancer types [227,228,229]. Capsaicin also hindered IL6-induced STAT3 activation. It impaired the advancement of multiple myeloma xenograft tumors in male thymic mice by suppressing STAT3 activation by inactivating JAK1 and c-Srcin in various preclinical cancer models [139] (Table 3).

#### 5.3.2. Evodiamine

Evodiamine, obtained from *Tetradium ruticarpum* (A. Juss.) T.G. Hartley., is a multitargeted anticancer agent [230]. In hepatocellular carcinoma cells (HepG2), evodiamine effectively hindered constitutive and IL-6-induced activation of STAT3 (Tyr705) phosphorylation. Evodiamine suppressed the phosphorylation of JAK2, Src, and extracellular signal-regulated kinases (ERK1/2). Additionally, evodiamine impeded STAT3–DNA binding activity [81]. The proliferation of HepG2 cells was inhibited by evodiamine via cell cycle arrest in the G2/M phase. It also downregulated the protein expression levels of cyclin D1, Bcl-2, Mcl-1, survivin, VEGF, XIAP, MMP-9, and HIF1-α in SMMC-7721 and HepG2 hepatocarcinoma cells [231].

#### 5.3.3. Indirubin

Indirubin, a naturally occurring alkaloid, is isolated from indigo dye-containing herbs (*Angelica sinensis* (Oliv.) Diels). It is an important potent component of “Danggui Longhui Wan”, a traditional Chinese medicine formulation used for the therapeutic intervention of inflammation, cancer, and various other chronic diseases. Recent research shows the potential of indirubin and related compounds in the treatment of cancer, especially inflammation-associated malignancies [232,233]. It is thought to hinder numerous kinases, and it might be utilized to treat chronic myelocytic leukemia, cancer, and neurodegenerative disorders. Indirubin has been shown to impede cyclin-dependent kinase dimers, triggering cell cycle arrest at G1/S or G2/M phase. Additionally, it repressed phosphorylated STAT3 expression [234]. 

### 5.4. Saponins

These components exist in diverse plants, and several alkaloids inhibit the JAK/STAT pathway as described below. Saponins are glycosidic secondary metabolites present in plants, grains, soybeans and certain herbs. Saponins may have one to three straight/branched chains of sugar (either D-glucose, D-xylose, D-galactose, L-arabinose L-rhamnose, or D-glucuronic acid). In vitro enzymatic assays and preclinical animal studies reveal that the saponins possess anticancer properties [235]. Interestingly, various saponins are known to augment the efficiency of several chemotherapeutic drugs, such as cisplatin, cyclophosphamide, docetaxel, doxorubicin, mitoxantrone, and paclitaxel [236]. The anticancer actions of saponins could be linked to their capability to inhibit the JAK/STAT pathway. 

#### β-Escin

β-Escin, a triterpene–saponin mixture, is extracted from the seeds of horse chestnut (*Aesculus hippocastanum* L.). β-Escin displayed anticancer action in various cancer models and increased the impact of chemotherapeutic agents [237]. β-Escin inhibited JAK1/2 and c-Src phosphorylation in HCC cells [238]. β-Escin also downregulated the expression levels of STAT3-regulated genes, for example, Bcl-2, Bcl-xL, cyclin D1, Mcl-1, survivin, and VEGF [238].

### 5.5. Steroids

Various plant-derived steroids, including phytosterols, possess anticarcinogenic properties [239], and the intake of high phytosterol was inversely related to an elevated cancer risk [240]. 

#### 5.5.1. Diosgenin

Diosgenin is a steroidal sapogenin that occurs in nature. It is a bioactive component present in fenugreek (*Trigonella foenum-graecum* L.) seeds. Studies show that diosgenin might confer several health benefits, including cancer preventive and anticancer effects [241,242,243]. Diosgenin suppressed the phosphorylation of *c*-Src, JAK1, and JAK2 in HCC cell lines (C3A and HepG2) [244]. This phytochemical increased the expression of SH-PTP2, and consequently inhibited STAT3 activation triggered by IL-6 [244]. It also hindered the proliferation of primary human thyrocytes by inducing apoptosis and causing cell cycle arrest in the G0/G1 phase [245].

#### 5.5.2. Ergosterol Peroxide

Ergosterol peroxide, derived from many plants and edible mushrooms, is known to possess immune-suppressive, antimicrobial, anti-inflammatory and antitumor effects [246]. It inhibited the phosphorylation of STAT3 and consequently reduced the expression level of phospho-STAT3 and CD-34 (an angiogenesis marker). Ergosterol peroxide also suppressed STAT3–DNA binding in multiple myeloma cells (U266). Moreover, it blocked phosphorylation of tyrosine kinases JAK2 and Src, but elevated the expression level of SHP-1 at the transcriptional level [80].

#### 5.5.3. Guggulsterone

Guggulsterone, an antagonist of the farnesoid X receptor, is a phytosterol present in the gum resin of guggul plant (*Commiphora mukul* (Hook. ex Stocks) Engl.). Guggulsterone is broadly utilized in Indian traditional medicine for the management of various diseases, including cancer [247,248,249]. It suppressed U266 cell proliferation by inducing apoptosis and cell cycle arrest at the sub-G_1_ phase. It decreased the levels of Bcl-xL, Bcl-2, cyclin D1, Mcl-1, and VEGF. It activated caspase-3 and cleavage of PARP protein. In particular, guggulsterone abrogated the stimulation of *c*-Src and JAK2 and therefore inactivates STAT3 in human multiple myeloma cells [250]. It also suppressed the activation of STAT3 triggered by IL-6. Guggulsterone also elevated the expression of SHP-1, a nontransmembrane protein and negative regulator of the JAK/STAT pathway [251].

### 5.6. Lignan

Lignans are present in an assortment of natural products, including pumpkin seeds, flax seeds, sesame seeds, broccoli, soybean, and various berries. Secoisolariciresinoldiglucoside, the primary lignan found in flax seeds, is metabolized in the colon to produce mammalian lignans, such as enterodiol and enterolactone. Once lignan is ingested, it suppresses the development of malignant tumors, specifically hormone-sensitive ones, such as neoplasms of the prostate, breast, and endometrium [252].

#### Arctiin

Arctiin, anarctigenin glucoside, is found in numerous herbs of Asteraceae family, especially *Centaurea imperialis* Hausskn. ex Bornm. and *Arctiumlappa* L. (greater burdock), and in *Forsythia viridissima* Lindl., *Himalaiella heteromalla* (D.Don) Raab-Straube, and *Trachelospermum asiaticum* (Siebold & Zucc.) Nakai. Arctiin drew the attention of natural product researchers due to its significant therapeutic benefits in relation to inflammation and malignancy [253]. In human multiple myeloma cells, it suppressed the constitutive activation of STAT3 phosphorylation at tyrosine 705 residue. Arctiin abrogated the constitutive activation of Src phosphorylation and the activation of JAK1/2. Moreover, treatment of U266 cells with arctiin elevated the level of protein tyrosine phosphatase ε (PTPε), and the silencing of PTPε produced an inversion of the arctiin-induced PTPε expression and the blockage of STAT3 phosphorylation. Arctiin additionally inhibited the proliferation of U266 carcinoma cell line via apoptosis induction and halting cells at G2/M phase [254].

### 5.7. Phytoalexin

Phytoalexins are produced in plants when they are attacked by invading organisms. They act as toxins and are formed when the host plants come in contact with parasites. Phytoalexins have been found in at least 75 plants, including soybeans, cruciferous vegetables, tomatoes, beans, rice, garlic, potatoes, and grapes. In preclinical studies, phytoalexins demonstrated anticancer actions by hindering proliferation, invasion and metastasis, hormonal stimulation, and modulatory effects on expression of xenobiotic-metabolizing enzymes [255].

#### Brassinin

Brassinin, a phytoalexin present in cruciferous vegetables, displayed antiproliferative, anticancer, and chemopreventive properties [256]. In A549 human lung carcinoma cells, it inhibited constitutive and IL-6-inducible STAT3 activation. Additionally, brassinin induced protein inhibitors of activated STATs-3 (PIAS-3) protein and mRNA, while SOCS-3 expression was diminished. Ablation of PIAS-3 by small interfering RNA inhibited brassinin-mediated cytotoxicity and the hindrance of STAT3. Suppressor of cytokine signaling-3 (SOCS-3), when overexpressed in brassinin-treated cells, increases phosphorylation of STAT3 and the viability of the cell. Brassinin downregulated the expression of STAT3-directed genes (phospho-STAT3, Ki-67, and CD31), suppressed proliferation and invasion, and induced apoptosis in the xenograft lung cancer (A549) mouse model. When administered intraperitoneally, the combination of paclitaxel and brassinin diminished tumor growth by the downregulation of phospho-STAT3, CD31, and Ki-67 in a xenograft lung cancer model [257].

## 6. Conclusions and Future Perspectives

Cancer is a dangerous health risk for people worldwide. The morbidity and the mortality rates connected with cancer are alarming, despite the existence of multiple treatment modalities for patients suffering from this disease. Natural substances are important components that can be used for the discovery and the progression of new anticancer medications. Consumption of dietary and medicinal plants that contain active phytoconstituents has minimal or negligible adverse effects. Natural components, when extracted, purified, concentrated, and administered at higher therapeutic doses, very often exhibit adverse side effects apart from their beneficial anticancer activities. Hence, the development of these phytoconstituents can only be progressed further following the confirmation of their safety based on toxicity studies in different preclinical animal models. Once a safety profile is established, the utilization of these substances to develop novel anticancer agents would provide a promising choice for chemoprevention and novel cancer treatment. The current therapeutic intervention for the management of cancer poses several limitations due to the use of monotargeted synthetic agents, elevated cost, low effectiveness, and dangerous adverse actions. Consequently, it is necessary to develop novel and innovative medications for prevention and treatment of cancer. Phytochemicals, including phenolics and polyphenols, terpenoids, alkaloids, saponins, and steroids, are known to prevent the development of many cancers. 

JAK/STAT is a vital signaling pathway implicated in the proliferation, differentiation, apoptosis evasion, and survival of neoplastic cells. In addition, its aberrant activation results in cancer-promoting mechanisms. This review indicates the capacity of several phytochemicals to act as multitargeted agents of JAK/STAT signaling inhibition in order to impede cancerous cell growth. Most of the phytochemicals examined in this review have been found to decrease the activity of JAK2 in addition to diminishing STAT3 activation (Figure 5). In addition, these phytochemicals reduce the expression of STAT3-regulated genes, namely VEGF, Bcl-2, Mcl-1, cyclin D, and survivin, and activate caspase-3, caspase-7, and caspase-9. Phenolics and polyphenols, terpenoids, alkaloids, saponins, and steroids present in natural sources inhibit the JAK/STAT signaling pathway and thereby inhibit cancer. The phenolic compounds obtained naturally include resveratrol, curcumin, EGCG, chalcone, caffeic acid, silibinin, butein, dihydroartemisinin, 5,7-dihydroxyflavone, honokiol, casticin, apigenin, and wedelolactone. All these phytoconstituents are efficient in arresting the progression of cancer cells. In addition, terpenoids, such as cucurbitacins (B, I, and Q), γ-tocotrienol, cryptotanshinone, celastrol, ursolic acid, pseudolaric acid B, oridonin, thymoquinone, parthenolide, and alantolactone, also have anticarcinogenic action against numerous malignant cells. Several alkaloids (such as evodiamine) and saponins (e.g., β-escin) also show antitumorigenic activity. Naturally occurring steroidal moieties (e.g., diosgenin, ergosterol peroxide, and guggulsterone) also exhibit anticancer activity. The aforementioned phytoconstituents act as negative regulators of the JAK/STAT pathway, which is a vital oncogenic signaling pathway for development and progression of cancer. Members of the JAK family are more commonly targeted than STATs by the negative regulators. JAK is STAT’s upstream protein, and therefore must be activated before STAT can be activated as well. Other signaling pathways that are independent of JAK can also reportedly activate the STATs [258]. Additionally, the anticancer effects of the various phytochemicals, as described in this review, may be achieved through a variety of complementary and multitargeted mechanisms, including modulation of yet unidentified targets, and one of these mechanisms could be the inhibition of the JAK/STAT signaling pathway.

Most of the current literature on the effects of bioactive phytoconstituents on JAK/STAT signaling in cancer is limited to in vitro studies. Further in vivo studies must be carried out to confirm the potential for bioactive constituents in suppressing and/or treating cancer by inhibiting the JAK/STAT signaling pathway. The limitation of many of these plant chemical components includes low aqueous solubility, limited absorption, and inadequate bioavailability, which may restrict their therapeutic use in clinical settings. It is indeed challenging to achieve sufficient concentration and/or bioavailability of a phytochemical via dietary consumption of its corresponding source food comparable to concentrations used in a research context. Human beings would need to eat an extraordinary amount of plant-based foods, which may not be feasible, to obtain the same results as achieved in in vitro or in vivo studies. However, it is also true that, unlike drugs, which may be administered at a specific dose and at a particular time under controlled conditions, phytochemicals are generally consumed at low and variable levels as a part of a complex dietary regimen at irregular intervals over a protracted period of time. Nevertheless, alternative or coadjuvant dietary interventions might help in the prevention and/or treatment of cancer.

Further research studies need to focus on investigation of the long-term impacts of phytochemical usage, the cause of the JAK/STAT pathway inhibition, the impact of phytochemicals to prevent cancer in high-risk populations, and their effects when used in combination with existing chemotherapy and when used in conjunction with diversified phytoconstituents. The impressive amount of research findings presented here establishes the promise of phytochemicals as anticancer agents and the necessity for implementation of phytochemicals in human clinical trials. Regardless, due to the side effects and resistance to current chemotherapeutic agents, phytochemicals must be further investigated in order to combat the cancer epidemic throughout the world.

Chemopreventive phytochemicals responsible for the inhibition of the JAK/STAT pathway have been discussed throughout this article. A coordinated effort to discover new targets to boost the capability to restore normal JAK/STAT signaling will result in future chemopreventive and anticancer drugs. The findings of the studies presented in this review may enable researchers to create novel and effective cancer prevention and therapeutic approaches.

## Figures and Tables

**Figure 1 cells-09-01451-f001:**
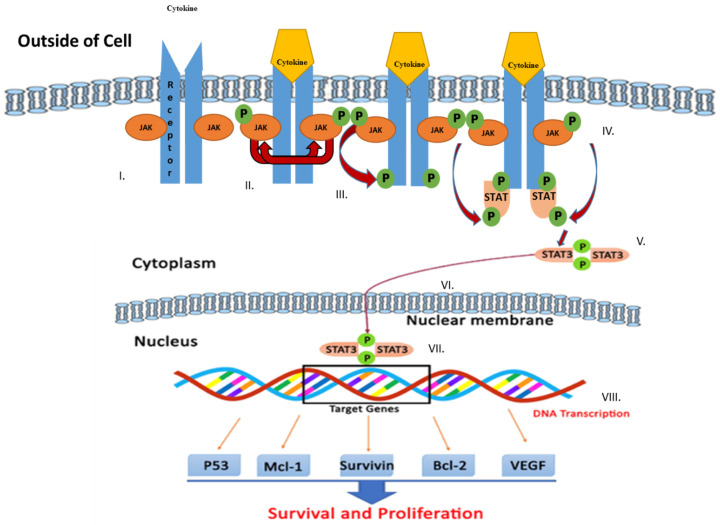
Schematic representation of Janus kinase (JAK)/signal transducer and activator of transcription (STAT) signaling pathway activation leading to upregulation of genes involved in survival and tumor cell proliferation. Various steps in the JAK/STAT pathway: I: Cytokine binds to receptor and receptors dimerize. II: JAKs are phosphorylated by each other. III: JAK phosphorylates the receptor, forming phosphotyrosine binding sites for STAT’s SH2 domain. IV: STAT binds to the receptor. JAK phosphorylates STAT which changes the conformation of STAT and stimulates its release. V: Phosphorylated STAT dissociates from the receptor and dimerizes. VI: Phosphorylated STAT translocates into the nucleus. VII. Phosphorylated STAT binds to DNA. VIII. Stimulation of DNA transcription of target genes.

**Figure 2 cells-09-01451-f002:**
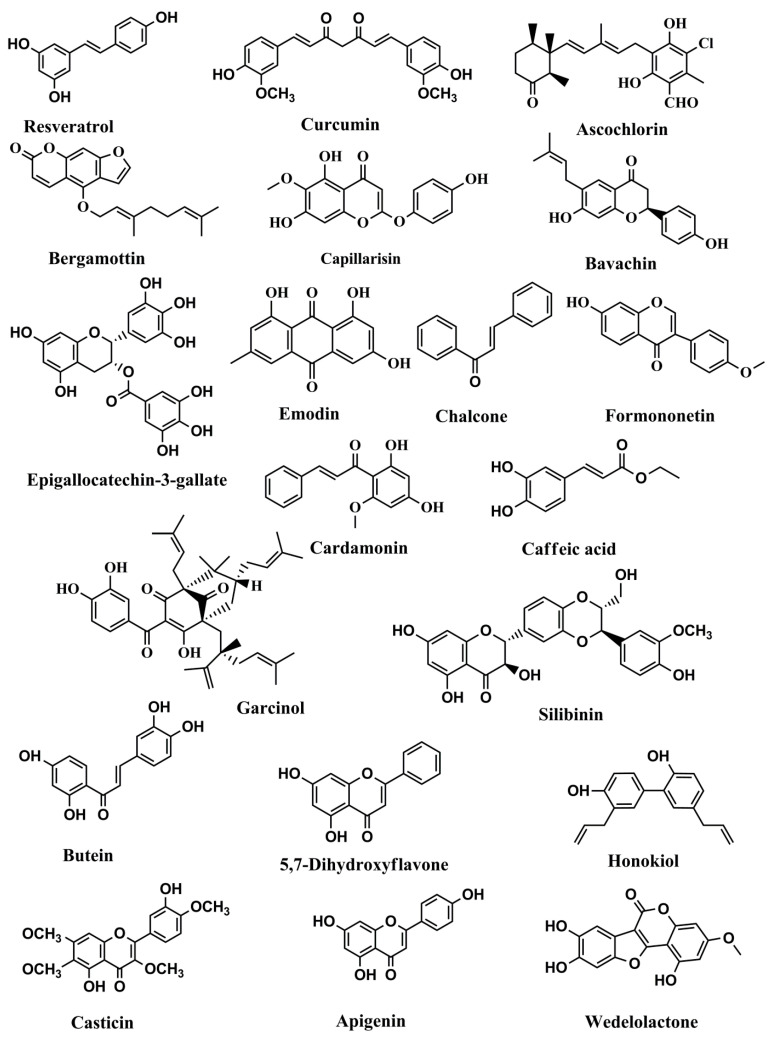
Structures of phenolics and polyphenols with anticancer activities correlated with inhibition of the JAK/STAT pathway.

**Figure 3 cells-09-01451-f003:**
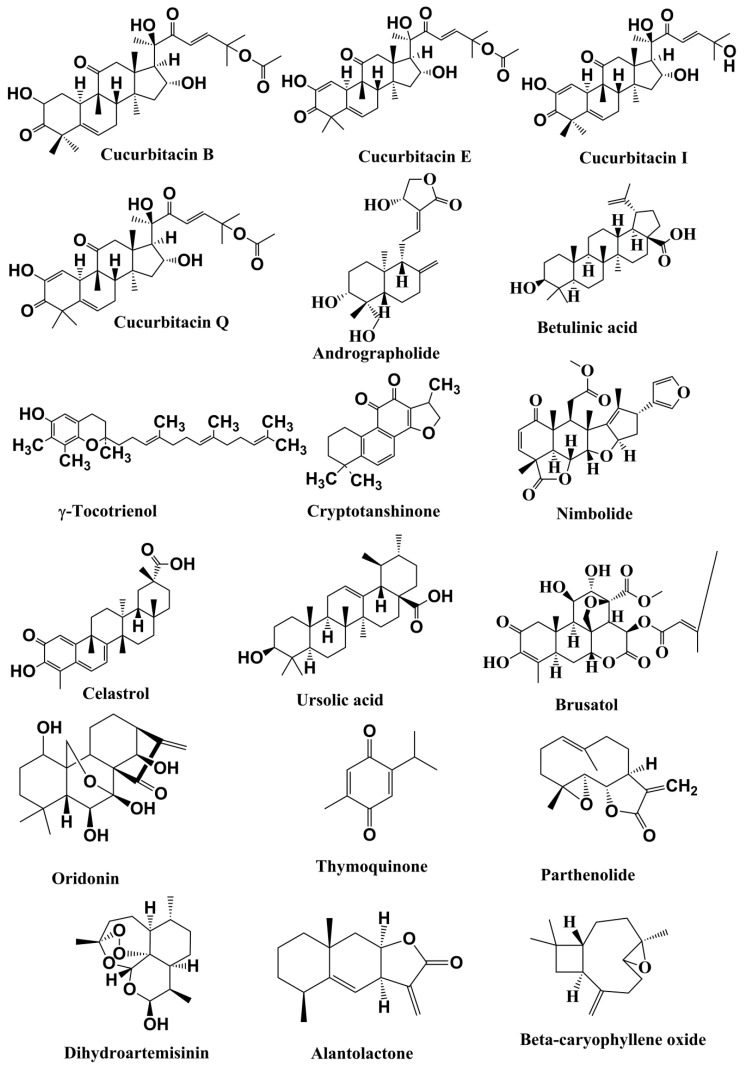
Structures of the terpenoids with anticancer activities related to inhibition of the JAK/STAT pathwa.

**Figure 4 cells-09-01451-f004:**
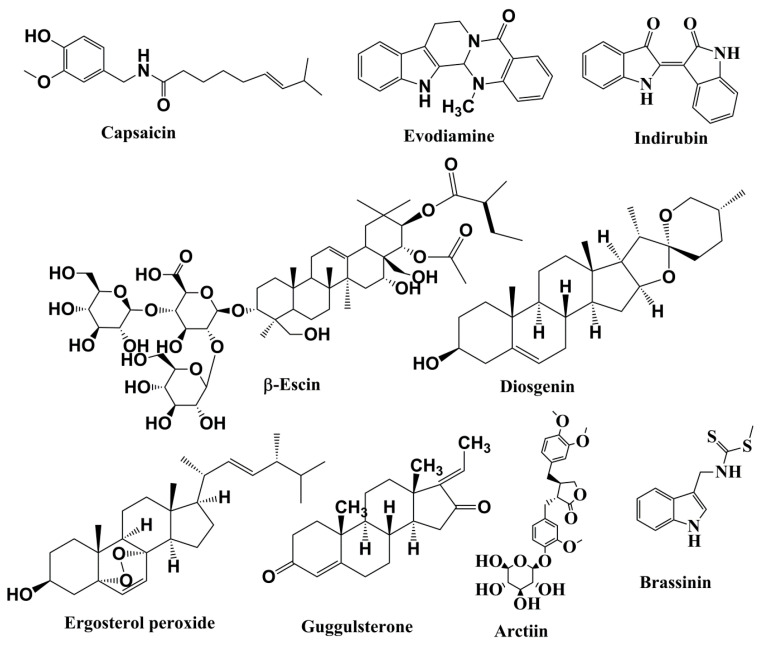
Structures of alkaloids, saponins, and steroids involved in anticancer activities related to inhibition of the JAK/STAT pathway.

**Figure 5 cells-09-01451-f005:**
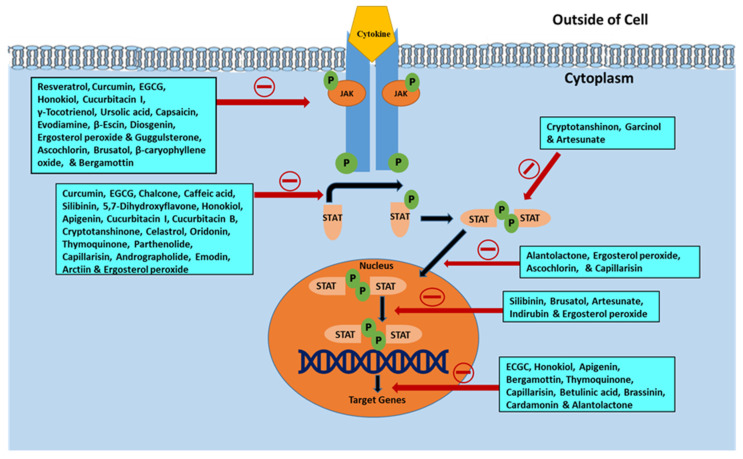
Illustration of various phytochemicals blocking specific steps in the JAK/STAT signaling pathway in relation to their cancer preventive and anticancer actions.

**Table 1 cells-09-01451-t001:** Potential anticancer activities of phenolics and polyphenols correlated with inhibition of the JAK/STAT signaling pathway.

Phytochemical Name	Sources	Anticancer Effect	Mechanism of Action	EC_50_/IC_50_	References
Resveratrol	*Vitis vinifera* L., *Vaccinium macrocarpon* Aiton and *Arachis hypogaea* L.	Inhibited proliferation in human epidermoid carcinoma (A431) cells	┴ Phosphorylation of JAK that prevented STAT1 phosphorylation	40 µM	Madan et al., 2008 [86]
		Inhibited proliferation in human multiple myeloma (U266 and RPMI 8226) cells	┴ Constitutive and inducible STAT3 activation	50 and 50 µM	Bhardwaj et al., 2007 [87]
		Inhibited tumor growth, induction of cytotoxicity, cell cycle arrest of v-Src-transformed mouse fibroblasts (NIH3T3/v-Src) at G0-G1 phase. Showed cytotoxicity in human breast cancer (MDA-MB-231), pancreatic carcinoma (Panc-1), and prostate carcinoma (DU145) cells	┴ Src tyrosine kinase activity; ┴ constitutive STAT3 activation	40, 70. and 25 µM	Kotha et al., 2006 [88]
Curcumin	*Curcuma longa* L.	Induction of cytotoxicity in human multiple myeloma (U266, RPMI 8226, and MM.1S) cells	┴ Constitutive and IL-6-inducible STAT3 phosphorylation; ┴ IFN-inducible STAT1 phosphorylation	13.8, 12.7 and 10 µM	Bharti et al., 2003 [95]
		Inhibited proliferation in human small cell lung cancer (NCI-H446 and NCI-1688) cells	┴ STAT3 phosphorylation; ↓ STAT3 downstream gene expression	15 and 15 µM	Yang et al., 2012 [96]
		Antitumor activity in T-cell leukemia, treatment causes growth arrest with increase in apoptosis	┴ Activation of the JAK/STAT pathway; ↓ JAK and STAT phosphorylation	Not specified	Rajasingh et al., 2006 [97]; Kunnumakkara et al., 2017 [98]
Ascochlorin	*Ascochyta viciae*	Inhibited cell migration and invasion in U373MG and A172 cancer cells	┴ MMP-9 expression, ┴ MMP-2 gelatinolytic action and expression; ┴ JAK2/STAT3 phosphorylation, cancer cell migration, and nuclear translocation of STAT3	10 μM	Cho et al., 2018 [100]
Bergamottin	Grape juice, lime, lemon, and bergamot oils	Antitumor activity in U266 cells, induction of substantial apoptosis at sub-G1 stage	┴ Constitutive STAT3 activation; ┴ Phosphorylation of JAK 1/2 and c-Src	100 μM	Kim et al., 2014 [102]
Capillarisin	*Artemisia capillaries* Thunb.	Negative regulator of growth and metastasis in human multiple myeloma cells; induces apoptosis; downregulated the expression level of various STAT3-regulated proteins	┴ Constitutive and inducible STAT3 activation at tyrosine 705; ┴ STAT3 phosphorylation; ┴ Activation of JAK1, JAK2, and c-Src kinases	Not mentioned	Lee et al., 2014 [103]
Bavachin	*Psoralea corylifolia* Linn. [*Cullen corylifolium* (L.) Medik.]	Induced cytotoxicity in multiple myeloma cell lines; induced apoptosis by activation of caspase-3 and caspase-9; inhibited activation of NF-κB, expression levels of Bcl-2, X-linked inhibitor of apoptosis protein (XIAP), survivin, and B cell lymphoma-extra large (Bcl-xL)	┴ Activation of STAT3; ┴ Phosphorylation of STAT3	10 μM	Takeda et al., 2018 [104]
Epigallocatechin-3-gallate (EGCG)	*Camellia sinensis* (L.) Kuntze	Anticancer activity in human pancreatic (AsPC-1 and PANC-1), breast (T47D), head and neck cancer (YCU-H861) cells	┴ Phosphorylation and expression of both JAK3 and STAT3 proteins	40, 40, 14.17 µM, 1 µg/mL	Tang et al., 2012 [109]
Emodin	*Rheum palmatum* L.	Stimulated the antiproliferation activity of interferon α/β in cervical carcinoma cell line (HeLa) and antitumor activity in Huh7 (hepatocellular cancer cell)-bearing mice in vivo	┴ STAT3 activation, ┴ 26S proteasome activity ↑ STAT1 phosphorylation, ┴ STAT3 phosphorylation	Not specified, 1.22 μM	He et al., 2016 [117]
Chalcone	*Malus domestica* (Suckow) Borkh. and *Solanum lycopersicum* L.	Induced cytotoxicity in Bovine aortic endothelial cells (BAEC)	┴ IL-6-induced and LPS-induced STAT3 phosphorylation	0.0069 µM	Liu et al., 2007 [120]
Formononetin	*Astragalus mongholicus* Bunge	Inhibited proliferation and invasion of colon carcinoma cell lines HCT116 and SW1116, cell cycle arrest at the G0/G1 stage	┴ STAT3 signaling pathway; ┴ STAT3 phosphorylation	Not specified	Wang et al., 2018 [123]
Garcinol	*Garcinia indica* (Thouars.) Choisy	Induced cytotoxicity in hepatocellular carcinoma cells, cell cycle arrest at G0/G1 phase	┴ Constitutive and IL-6 inducible STAT3 activation	Not specified	Sethi et al., 2014 [126]
Cardamonin	*Alpinia conchigera* Griff., and *Alpinia hainanensis* K. Schum.	Antitumor activity in U87 cells (in vitro) and CD133+ GSCs (in vitro and in vivo); induced apoptosis	┴ STAT3 signaling pathway; ┴ STAT3 activation; ┴ downstream STAT3 gene expressions (i.e., Bcl-xL, Bcl-2, Mcl-1, survivin, and VEGF)	Not specified	Wu et al., 2015 [128]
Caffeic acid or its derivative 3-(3,4-dihydroxy-phenyl)-acrylic acid 2-(3,4-dihydroxy-phenyl)-ethyl ester (CADPE)	*Camellia sinensis* L. Kuntze., and *Coffea arabica* L.	Induction of cytotoxicity in human renal carcinoma (Caki-1) cells	┴ STAT3 phosphorylation; ┴ Src tyrosine kinase.	30 µM	Jung et al., 2007 [131]
Silibinin	*Silybum marianum* (L.) Gaertn.	Induction of cytotoxicity in human prostate cancer (DU145) cells	┴ Constitutively active STAT3, ↑ apoptosis and ┴ constitutive STAT3–DNA binding	50 µM	Agarwal et al., 2007 [135]
		Suppressed transcriptional function in urethane-induced lung tumors in A/J mice	┴ STAT3 phosphorylation	Not specified	Tyagi et al., 2009 [136]
Butein	*Toxicodendron vernicifluum* (Stokes) F.A. Barkley and *Butea monosperma* (Lam.) Kuntze	Exhibited antitumor activity in human hepatocellular carcinoma (HepG2 and SNU-387)	┴ Constitutive and IL-6- induced STAT3 activation by inactivating JAK1 and c-Src.	50 and 50 µM	Bhutani et al., 2007 [139]; Rajendran et al., 2011 [140]
5,7-Dihydroxyflavone	*Passiflora incarnate* L., *Passiflora caerulea* L., and *Oroxylum indicum* (L.) Kurz	Inhibited proliferation HepG2 tumor xenografts in vivo	↓ Phosphorylation of STAT3	20 µM	Zhang et al., 2013 [143]
Honokiol	*Magnolia officinalis* Rehder & E. H. Wilson and *Magnolia grandiflora* L.	Inhibited proliferation, induced apoptosis in human leukemic cell lines (HEL and THP1), multiple myeloma cells (U266) and murine myeloid cell (32D)	┴ Constitutive and inducible STAT3 activation; ┴ mRNA levels of STAT3 target genes in a concentration-dependent manner; ↓ Nuclear translocation of STAT3	40 µM	Bi et al., 2015 [149]
Casticin	*Achillea millefolium* L., *Artemisia abrotanum* L., *Vitex trifolia* subsp. *Litoralis Steenis*, *Camellia sinensis* (L.) Kuntz, *Centipeda minima* (L.) A.Braun & Asch, *Clausena excavate* Burm.f., *Crataegus pinnatifida* Bunge, *Croton betulaster* Müll Arg., *Daphne genkwa* Siebold & Zucc., *Ficus microcarpa* L.f., *Nelsonia canescens* (Lam.) Spreng., *Pavetta crassipes* K. Schum., *Vitex trifolia* subsp. *Litoralis steenis*, *Vitex agnus-castus* L., *Vitex negundo* L. and *Vitex trifolia* L.	Inhibited proliferation, induced apoptosis, cell cycle arrest at G2/M phase in colon (Panc-1), breast (MCF-7), lung (A549), gastric (SGC-7901), ovarian (SKOV3), liver (HepG2), leukemia (K562) cancer cells	┴ Constitutively active STAT3 and modulates STAT3 activation by modifying upstream STAT3 regulator activity.	10, 8.5, 14.3, 1, 2.18, 30, 5.95 µM	Chen et al., 2011 [152]; Zeng et al., 2012 [153]
		Induced apoptosis in 786-O, YD-8, and HN-9 cancer cells	┴ constitutively activation of STAT3; Modulated STAT3 activation by altering the activity of upstream STAT3 regulators, and abrogated IL-6-induced STAT3 activation; ┴ JAK/STAT pathway	5 μM	Lee et al., 2019 [154]
Apigenin	*Petroselinum crispum* (Mill.) Fuss., *Apium graveolens* L., and *Matricaria chamomilla* L.	Anticancer and antitumor activity in colon cancer (HCT-116) cells	┴ Phosphorylation of STAT3 and consequently downregulated the antiapoptotic proteins Bcl-xL and Mcl-1	47.33 µM	Ozbey et al., 2018 [25]; Maeda et al., 2018 [157]
		Anticancer activity in BT-474 (breast cancer) cells	┴ JAK/STAT pathway ┴ STAT3 nuclear accumulation ┴ Phosphorylation of JAK1/2, and STAT3	Not specified	Ozbey et al., 2018 [25]
Wedelolactone	*Eclipta prostrate* (L.) L. and *Sphagneticola calendulacea* (L.) Pruski	Inhibited proliferation, induced apoptosis, causes cell cycle arrest at S and G2/M phases in breast (MDA-MB-231) and HepG2 cancer cells	┴ STAT1 dephosphorylation and prolonging STAT1 activation, ┴ T-cell protein tyrosine phosphatase	Not specified in MDA-MB-231; 20 µM (EC_50_) in HepG2 cells	Benes et al., 2011 [159]; Benes et al., 2012 [160]; Chen et al., 2013 [161]

Various symbols (↑, ↓ and ┴) indicate increase, decrease and inhibition in the obtained variables, respectively.

**Table 2 cells-09-01451-t002:** Potential anticancer activities of terpenoids related to inhibition of the JAK/STAT signaling pathway.

Phytochemical Name	Sources	Anticancer Effect	Mechanism of Action	EC_50_ /IC_50_	References
Cucurbitacin B	*Hemsleya endecaphylla* C.Y. Wu	Induced cytotoxicity in human pancreatic cancer (MiaPaCa-2, AsPC-1) cells	┴ JAK2, ┴ STAT3, and ┴ STAT5 activation	0.278, 0.017 μM	Thoennissen et al., 2009 [168]; Zhou et al., 2017 [169]
		Induced cytotoxicity in leukemia K562 cells	┴ STAT3 activation	50 µM	Chan et al., 2010 [170]
Cucurbitacin E	*Wilbrandia ebracteate* Cogn.	Induced cytotoxicity in human bladder cancer (T24) cells	↓ Levels of phosphorylated STAT3 (p-STAT3)	1012 nM	Huang et al., 2012 [171]
Cucurbitacin I	*Cucumis melo* L.	Induced cytotoxicity in human lung adenocarcinoma (A549) cells	↓ Phosphotyrosine STAT3, ↓ JAK levels. ┴ STAT3-DNA binding; ┴ STAT3-mediated gene transcription	500 nM	Blaskovich et al., 2003 [172]
		Induced cytotoxicity in glioblastoma multiforme cells, G2/M cell cycle arrest by downregulating cyclin B1 and cdc2 expression	┴ JAK/STAT pathway; ↓ Phosphorylated STAT3 levels	Not specified	Su et al., 2014 [173]
		Inhibitory activity in Sézary (Sz) syndrome and anaplastic large cell lymphoma	┴ JAK/STAT pathway; ↓ Phosphorylated STAT3 levels	30 μM	van Kester et al., 2008 [174]; Shi et al., 2006 [175]
Cucurbitacin Q	*Picrorhiza kurrooa* Royle ex Benth.	Antiproliferative effect in human non-small-cell lung carcinoma (A549) cells	↓ STAT3 but not JAK2 activation	3.7 µM	Sun et al., 2005 [176]
Andrographolide	*Andrographis paniculata* (Burm.f.) Nees.	Enhanced anticancer activity of doxorubicin	┴ STAT3 signaling pathway; ┴ Constitutively actuated and IL-6-induced phosphorylation of STAT3 and subsequent nuclear translocation	Not specified	Zhou et al., 2010 [179]
Betulinic acid	White-barked birch plants	Inhibited proliferation and induced apoptosis in human multiple myeloma (U266) cells; Cell cycle arrest at sub-G1 stage; downregulated the expression level of STAT3-regulated gene products	┴ STAT3 signaling, ┴ STAT3-directed gene expression; ┴ Constitutive activation of STAT3, Src kinase, and JAK1/2	Not specified	Pandey et al., 2010 [182]
*γ*-Tocotrienol	*Elaeis guineensis* Jacq.	Anticancer effect in human hepatocellular carcinoma (HepG2, C3A, Hep3B, SNU-387, and PLC/PRF5) cells	┴ Both constitutive and inducible activation of STAT3; ┴ Phosphorylation of JAK1, JAK2, and c-Src	Not specified	Rajendran et al., 2011 [185]; Banerjee et al., 2019 [186]
Cryptotanshinone	*Salvia miltiorrhiza* Bunge	Induced cytotoxicity in human prostate cancer (DU145) cells	┴ Phosphorylation of STAT3 through an independent mechanism of JAK2 phosphorylation	7.59 µM	Shin et al., 2009 [188]
Nimbolide	*Azadirachta indica* A. Juss.	Induced cytotoxicity and cell cycle arrest at G1–S stage in glioblastoma multiforme cells. It downregulates Bcl2	┴ STAT3 pathway, ┴ STAT3 phosphorylation	Not specified	Karkare et al., 2014 [191]; Zhang et al., 2016 [192]
Celastrol	*Tripterygium wilfordii* Hook.f.	Induced cytotoxicity in human multiple myeloma (U266, RPMI 8226 and RPMI-8226-LR-5) cells	┴ Phosphorylation of STAT3 as well as STAT3-mediated IL-17 expression	Not specified	Kannaiyan et al., 2011 [193]
		Inhibited differentiation and cell proliferation in multiple myeloma cells	┴ STAT3 phosphorylation; ┴ STAT3-mediated IL-17 expression ┴ T-helper 17 (Th17)	Not specified	Banerjee et al., 2019 [186]
Ursolic acid	*Mirabilis jalapa* L.	Induced cytotoxicity in human prostate cancer (DU145 and LNCaP) cells	┴ Activation of constitutive and inducible STAT3; ↓ phosphorylation of Src and JAK2	80 and 47 µM	Shanmugam et al., 2011b [198]
		Inhibited tumor growth in prostate xenograft tumor in TRAMPmice in vivo	┴ JAK/STAT signaling; ┴ activation of STAT3	Not specified	Shanmugam et al., 2011 [198]
Brusatol	*Brucea javanica* (L.) Merr.	Induced cytotoxicity in head and neck squamous cell carcinoma	┴ STAT3 signaling pathway; ┴ Activation of STAT3 as well as JAK1/2, and Src	Not specified	Lee et al., 2019 [199]
Oridonin	*Isodon rubescens* (Hemsl.) H. Hara	Inhibited proliferation and induced apoptosis and cell cycle arrest at G2/M phase in breast (MCF-7), leukemia (K562), lung (A549), prostate (PC-3), liver (Bel7402), gastric (BGC823), and uterine cervix cancer (HeLa) cells The viability of BxPC-3 (human pancreatic cancer) cells is reduced on treatment	┴ STAT3 signaling pathway; ↓ IL-6, ↓ STAT3, and ↓ phospho-STAT3 expression level, ↓ p21, ↓ cyclin A, ↓ cyclin B1, ↓ cyclin D1, ↓ VEGF, and ↓ MMP-2	18.4, 4.3, 18.6, 15.2, 7.6, 13.7 μg/mL 19.32 μg/mL	Bu et al., 2012 [202]; Chen et al., 2008 [203]; Chen et al., 2012 [204]; Chen et al., 2014 [205]
Thymoquinone	*Nigella sativa* L.	Induced cytotoxicity in multiple myeloma (U266 and RPMI 8226) cells.	┴ STAT3 phosphorylation	15 and 15 µM	Zhu et al., 2016 [208]; Li et al., 2010 [209]
Parthenolide	*Tanacetum parthenium* (L.) Sch. Bip., *Tanacetum vulgare* L., *Centaurea ainetensis* Boiss., *Tanacetum larvatum* (Pant.) Hayek., and *Helianthus annuus* L.	┴ Proliferation, ↑ apoptosis, cell cycle arrest at G2/M phase in breast (MCF-7), skin (MDMB-231), melanoma (LCC9), malignant glioma (ABCB5+), epidermal tumorigenesis (A375), liver (1205Lu), gastric cancer (WM793) cells	┴ IL-6-induced STAT3 phosphorylation; ┴ JAK2 kinase activity	9.54, 10 µM, 600 nM, 12, 2.9, 6, 12 µM	Cheng et al., 2011 [211]; Liu &Xie, 2010 [212]; Shanmugam et al., 2011 [213]
		Inhibited JAK2 kinase activity in MDA-MB-231 cells	┴ IL-6-induced STAT3 phosphorylation	Not specified	Liu et al., 2018 [214]
Dihydroartemisinin	*Artemisia annua* L.	Inhibited tumor cell growth in human head and neck cancer (FaDu), liver cancer (Hep-G2), colon cancer (HCT-116), and tongue cancer (Cal-27) cells	┴ JAK2/STAT3 signaling activation; ↓ targeted proteins	160, 80, 25, 80 µM	Jia et al., 2016 [217]; Wang et al., 2017 [218]
Alantolactone	*Aucklandia costus* Falc., *Inula helenium* L., *Inula japonica* Thunb., and *Inula racemose* Hook. f.	Antiproliferative effect in glioblastoma (U87), colon (HCT-8), leukemia (HL-60), liver (HepG2), lung cancer (A549) cells	┴ Both constitutive and inducible STAT3 activation at tyrosine 705; ┴ STAT3 translocation to the nucleus; ┴ DNA-binding, and ┴ STAT3 target gene expression	135.27 µM, 5 µg/mL, 1.1, 40, 8.39 µM	Pal et al., 2010 [220]; Khan et al., 2012 [221]; Chun et al., 2015 [222]
*β*-Caryophyllene oxide	*Cannabis sativa* L., *Humulus lupulus* L., *Origanum vulgare* L., *Psidium guajava* L., *Salvia Rosmarinus* Spenn., and *Syzygium aromaticum* (L.) Merr. & L. M. Perry	Inhibited proliferation in multiple myeloma, prostate, and breast cancer cell lines	┴ STAT3 pathway; ┴ STAT3 activation as well as JAK 1/2 and c-Src	Not specified	Kim et al., 2014 [224]

Various symbols (↑, ↓ and ┴) indicate increase, decrease and inhibition in the obtained variables, respectively.

**Table 3 cells-09-01451-t003:** Anticancer activities of alkaloids, saponins, steroids, lignan, and phytoalexin related to inhibition of the JAK/STAT signaling pathway.

Phytochemical Class	Phytochemical Name	Sources	Anticancer Effect	Mechanism of Action	EC_50_/IC_50_	References
Alkaloids	Capsaicin	*Capsicum frutescens* L.	Induction of cytotoxicity in human multiple myeloma (U266 and MM.1S) cells	┴ Constitutive and IL-6-induced activation of STAT3; ┴ JAK1 and c-Src activation	50 and 50 µM	Bhutani et al., 2007 [139]
	Evodiamine	*Tetradium ruticarpum* (A. Juss.) T. G. Hartley	Inhibited proliferation, and induced apoptosis; Cell cycle arrest at G2/M phase in murine Lewis lung (LLC), hepatocellular (HepG2), leukemia (K562), gastric (SGC-7901), colon (COLO-205) cancer cells.	↓ Constitutive and IL-6-induced activation of STAT3 tyrosine 705 (Tyr705); ↓ JAK2, Src and ERK1/2; ┴ STAT3–DNA binding activity	113, 8.516, 5, 10, 27.15 µM	Yang et al., 2013 [81]
	Indirubin	*Angelica sinensis* (Oliv.) Diels	Reduced cell viability in human prostate and breast cancer cells; induced apoptosis; Cell cycle arrest at G1/S or G2/M	┴ STAT3 signaling; ┴ STAT3 phosphorylation	~4 µM for each cell line	Chen et al., 2018 [234]
Saponins	β-Escin	*Aesculus hippocastanum* L.	Induction of cytotoxicity in human hepatocellular carcinoma (HepG2, PLC/PRF5, and HUH-7) cells	┴ Activation of STAT3 and induced expression of SHP-1; ┴ Phosphorylation of JAK1, JAK2, and c-Src	Not specified	Tan et al., 2010 [238]
Steroid	Diosgenin	*Trigonella foenum-graecum* L.	Induced cytotoxicity in human hepatocellular carcinoma (C3A, HUH-7, and HepG2) cells	┴ Constitutive and inducible activation of STAT3	100, 100 and 50 µM	Li et al., 2010 [244]
	Ergosterol peroxide	*Agaricus bisporus*	Induced cytotoxicity in human multiple myeloma (U266) cells	┴ Phosphorylation of JAK2; ┴ phosphorylation; ┴ DNA binding activity; ┴ Nuclear translocation of STAT3	Not specified	Rhee et al., 2012 [80]
	Guggulsterone	*Commiphora mukul* (Hook. ex Stocks) Engl.	Induced cytotoxicity in human multiple myeloma (U266) cells	┴ Constitutive andIL-6-induced STAT3; ┴ Phosphorylation of JAK2 and Src	25 µM	Ahn et al., 2008 [250]
Lignan	Arctiin	*Centaurea imperialis* Hausskn. ex Bornm., *Arctium lappa* L., *Forsythia viridissima* Lindl., *Himalaiella heteromalla* (D. Don) Raab-Straube., and *Trachelospermumas iaticum* (Siebold & Zucc.) Nakai	Inhibited proliferation, cell cycle arrest at G2/M stage in human multiple myeloma cells	┴ STAT3 phosphorylation in tyrosine 705; ┴ Constitutive enactment of Src phosphorylation and JAKs 1/2; ↑ mRNA, ↑ protein levels of protein tyrosine phosphatase ε; ┴ STAT3 regulated gene products	Not specified	Lee et al., 2019b [254]
Phytoalexin	Brassinin	*Brassica rapa* L.	Induced apoptosis in lung cancer cells (A549) in an in vivo mouse model	┴ STAT3 activation; ┴ Both constitutive and IL-6-inducible STAT3 activation; ┴ Phospho-STAT3, ┴ Ki-67 and ┴ CD31	Not specified	Lee et al., 2015 [257]

Various symbols (↑, ↓ and ┴) indicate increase, decrease and inhibition in the obtained variables, respectively.

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
