# Peer review of "Targeting the JAK/STAT Signaling Pathway Using Phytocompounds for Cancer Prevention and Therapy"

_cells, 2020, doi:10.3390/cells9061451_

Round 1

Reviewer 1 Report

This review discusses phytocompounds in cancer prevention and potential therapy by inhibiting JAK-STAT signaling. While it is well structured and quite comprehensive, further editing and proofreading is necessary.  For example, on page 5 line 202, "myeloma growth promoter" should be myeloma growth inhibitor. 

Author Response

The authors of this manuscript express their sincere thanks to the reviewer for the critical assessment of this work. The authors have acted upon the recommendations of the reviewer which have resulted in a significant enhancement of the quality of this manuscript. All modifications incorporated in the manuscript are highlighted using red color font. A “point-by-point” response to each and every comment is outlined below.

Comment:

This review discusses phytocompounds in cancer prevention and potential therapy by inhibiting JAK-STAT signaling. While it is well structured and quite comprehensive, further editing and proofreading is necessary.  For example, on page 5 line 202, "myeloma growth promoter" should be myeloma growth inhibitor.

Response:

We are deeply encouraged by the reviewer’s generous comments about the quality of our work. We sincerely apologize for the inadvertent errors. We have extensively edited our manuscript to limit typographic and grammatical errors. We have changed “myeloma growth promoter” to “myeloma growth inhibitor” (page 5, line 204).

On behalf of my co-authors, I once again express my sincere thanks to the erudite reviewer for the valuable suggestions and constructive input to improve the quality of our manuscript.

Reviewer 2 Report

In their review article titled "Targeting JAK/STAT signaling pathway using phytocompounds for cancer prevention and therapy" the authors provide an extensive and impressive review of the literature on the effects that several phytochemical groups have on the Jak/STAT pathway and tumorigenesis.
The manuscript is organized well, generally well written, and it could quickly become a prime reference for anyone trying to further research on targeting the STAT pathway as a therapeutic strategy (which is the alleged intent of the article). I would therefore like to begin by extending my congratulations to the authors for their manuscript.

However, I would also recommend some minor revisions before the manuscript can be accepted for publication. The following are some suggestions that I would hope might help the authors improve their manuscript.

Major points

1) Through their very exhaustive review of the literature, the authors provide plenty of examples of purified phytochemicals that have shown anti-tumorigenic effects in vitro and in some pre-clinical animal models of cancer. However, where is the evidence that dietary consumption of their corresponding source vegetables would achieve a concentration and/or bioavailability of these phytocompounds that is comparable to those that were used in a research context? Conversely, do we know if the physiological concentration achieved by consumption of the various vegetables mentioned throughout the article has any effect on the same experimental models? In the introduction to the article (L57-77), the authors correctly point out that alternative or co-adjuvant dietary interventions might help in the treatment or prevention of cancer, and point out to the importance of better understanding how natural compounds might impact some of the molecular pathways normally involved in cancer initiation and progression. Agreed. But almost none of the work reviewed in this manuscript connects the effects of a purified phytocompound observed in vitro to the amount of the same phytocompound that becomes truly incorporated from consuming their natural sources (or simple extracts as dietary supplements). I understand that none of that evidence might actually exist, so this is not blaming the authors for failing to include this information in their review. Instead, I would strongly recommend that the authors incorporate a paragraph or two to more explicitly acknowledge the various sources of disconnect that can exist between knowing that a purified phytocompound has an effect on cells in culture, and the therapeutic and/or preventive effect of consuming their natural source. After all, so many things can fail from the food plate to the cell culture dish.

2) Several of the compounds reviewed in this manuscript seem to act on the Jak/STAT pathway indirectly (e.g. through Src or PI3K). How specific is the requirement for the STAT pathway for their anti-tumorigenic effect? We simply don't know. It is possible that these compounds affect multiple molecular pathways, and that Jak/STAT inhibition only partially contributes to their therapeutic potential. Therefore, the authors should more explicitly discuss that many of these natural compounds may act molecularly on as of yet unidentified targets and have indirect effects on several pathways that collaboratively contribute to malignancy. For instance, in the legends to Figure 2 and Figure 3, the authors say "... anticancer activities mediated through inhibition of (the) Jak/STAT pathway..." If the original research showed the appropriate molecular epistasis experiments to demonstrate that these compounds indeed depended on Jak/STAT inhibition to exert their anti-tumorigenic effect, then the authors should summarize the information accordingly. If, on the other hand, the original pre-clinical studies only showed a correlation between tumor growth suppression and Jak/STAT inhibition, then all statements indicating that a given compound acted *via* Jak/STAT inhibition should be revised to indicate that the anti-tumorigenic effect correlated with inhibition of the pathway.

3) There is a strong emphasis throughout the manuscript on the therapeutic/medicinal effects of many natural compounds. It is perhaps stated most strongly in the Introduction (L62-66, and refs 8-14). Since the main premise of this review is to provide a potential molecular basis to the effectiveness of alternative medicines, the manuscript might benefit from a concise but more detailed description of the evidence of herbal medicines and dietary supplements with proven therapeutic or preventive effects on cancer treatment. In other words, if this review seeks to convey the idea that herbal medicine may function through its effect on known pathways related to cancer progression, the message would be supported by a more explicit description of the studies that provided sound evidence that herbal treatments have a demonstrable effect on the prevention or treatment of specific cancer types.

4) Section 3 (JAK/STAT signaling pathway) requires a thorough and careful revision. There is a lot of repetition (the pathway is described several times in different paragraphs), it often lacks logical order in the way that the pathway stages and components are described and it contains several phrases and terms that are either out of place or seem forcefully edited from other sources. For instance: "the improvement of the fetus" (L130), "receptor of the epidermal growth factor" (for Epidermal Growth Factor Receptor, or EGFR, L135), "the uplift (...) of cancer" or "chronic leukemia cells of the lymphoid" (for chronic lymphoblastic leukemia, L217). All of the previous seem forced attempts at changing the conventional terminology used to describe specific processes or molecules. While it is appreciated that the authors would want to re-phrase statements that they found in their review of the literature to avoid copying and pasting, they also need to be careful about respecting certain nomenclature conventions.

5) There are many cell lines that are mentioned by name throughout the article, but there's no mention to what type of cell line they are. Some examples include A172 and U373MG (L20, p13), U266 (L29, p13), THP1 and HEL (L184, p16) and others. I would recommend that the authors peruse their manuscript carefully to make sure that they indicate the cell types associated with cell lines mentioned, for those who are not experts and familiar with a specific cell line.

6) L90, p32: "Since natural components...cancer treatment" I beg to differ with the authors on this statement on two levels: 1) if natural components may not have adverse most often; but they usually do: we call them natural poisonous plants; 2) natural compounds present in low amounts in natural sources may not have noxious effects. But they can turn very toxic when extracted, purified and concentrated for use in therapy. For instance: taxol. If what the authors meant is that the consumption of natural herbs, plants and fungi that contain active phytocompounds will be rarely toxic, then it should be made more clearly explicit. Natural components that are given at higher (therapeutic) doses very often cause adverse side effects. I would therefore invite the authors to revise this sentence.

Minor points

L41-43: What do the authors mean by cancers being "dominated" by viral or bacterial infection and genetic mutations? What else could cause cancer?

L56-57: The authors should consider changing "...perceived as potential mechanisms of cancer growth..." to "...potential approaches to cancer treatment..." It makes better sense in the context of targeting and inhibiting the hallmarks of cancer progression.

L67-77: This paragraph is poorly organized - it leads with the potential mechanisms by which bioactive phytocompounds might impact cancer progression or treatment, and ends with broader statements about their potential. Perhaps the order of statements should be revised/reversed.

L78: consider "...initiating CHANGES in gene expression..."

L83: This sentence is confusing. Are the authors implying that "Inhibiting the JAK/STAT signaling pathway may [suppress tumorigenesis] by preventing [apoptosis]"? If inhibiting JAK/STAT is anti-apoptotic, then inhibiting JAK/STAT would be PRO-tumorigenic. This sentence needs revising.

L108: What is meant by "...STAT retained residues..."? Please clarify/revise.

L110-111: Consider "...DNA and stimulates THE EXPRESSION OF GENES that are responsive to cytokines..."

L120: Nonsensical sentence - please revise.

L144 (Fig 1 legend): Consider "...activation leading to up regulation of survival and tumor cell proliferation GENES..."

L155-159: Refs 48 and 49 do not seem to address the inhibition of inflammatory pathways by natural compounds (at leas by title) - please correct, or clarify further.

L204-205: "...has identified domains or residues whole functions to be delineated..." Is this a broken sentence? Please revise/edit.

L208-209: The meaning of "The canonical signaling is a characteristic of tyrosine phosphorylated STAT functioning that serve as transcription factors" is implicit, but the phrase needs rewriting.

L31, p13: delete "and" in "...by siRNA and abrogated..."

L66, p14: "various" should be changed to "two".

L91-93, p14: I find this statement confusing: a) How are the IFN-a and JAK/STAT pathways related to each other again? (the authors could summarize in 1-2 sentences what is known about their connection); b) if emodin inhibits STAT3 phosphorylation, would it act by "stimulating" (as written) or by INHIBITING the JAK/STAT signaling pathway. It's confusing.

L85-87, p32: "This section...can be drawn" looks like boilerplate language provided by MDPI as guidance for the authors. It should be taken out.

Author Response

The authors of this manuscript express their sincere thanks to the reviewer for the critical assessment of this work. The authors have acted upon the recommendations of the reviewer which have resulted in a significant enhancement of the quality of this manuscript. All modifications incorporated in the manuscript are highlighted using red color font. A “point-by-point” response to each and every comment is outlined below.

General comments:

In their review article titled "Targeting JAK/STAT signaling pathway using phytocompounds for cancer prevention and therapy" the authors provide an extensive and impressive review of the literature on the effects that several phytochemical groups have on the Jak/STAT pathway and tumorigenesis.

The manuscript is organized well, generally well written, and it could quickly become a prime reference for anyone trying to further research on targeting the STAT pathway as a therapeutic strategy (which is the alleged intent of the article). I would therefore like to begin by extending my congratulations to the authors for their manuscript.

However, I would also recommend some minor revisions before the manuscript can be accepted for publication. The following are some suggestions that I would hope might help the authors improve their manuscript.

Response:

We are also honored by the reviwer’s appreciation of our work and the recommendation for publication. As described below, we have addressed the reviewer’s specific comments and made the necessary changes to improve the quality of our manuscript.

Specific comments:

Major points

Comment 1:

Through their very exhaustive review of the literature, the authors provide plenty of examples of purified phytochemicals that have shown anti-tumorigenic effects in vitro and in some pre-clinical animal models of cancer. However, where is the evidence that dietary consumption of their corresponding source vegetables would achieve a concentration and/or bioavailability of these phytocompounds that is comparable to those that were used in a research context? Conversely, do we know if the physiological concentration achieved by consumption of the various vegetables mentioned throughout the article has any effect on the same experimental models? In the introduction to the article (L57-77), the authors correctly point out that alternative or co-adjuvant dietary interventions might help in the treatment or prevention of cancer, and point out to the importance of better understanding how natural compounds might impact some of the molecular pathways normally involved in cancer initiation and progression. Agreed. But almost none of the work reviewed in this manuscript connects the effects of a purified phytocompound observed in vitro to the amount of the same phytocompound that becomes truly incorporated from consuming their natural sources (or simple extracts as dietary supplements). I understand that none of that evidence might actually exist, so this is not blaming the authors for failing to include this information in their review. Instead, I would strongly recommend that the authors incorporate a paragraph or two to more explicitly acknowledge the various sources of disconnect that can exist between knowing that a purified phytocompound has an effect on cells in culture, and the therapeutic and/or preventive effect of consuming their natural source. After all, so many things can fail from the food plate to the cell culture dish.

Response:

We admire the reviewers for these thought-provoking comments. Many issues raised by the reviewer represent the fundamental challenges of natural product research. A detailed discussion on these issues is outside the scope of this review. However, as per the recommendation of the reviewer, we have addd a paragraph in the conclusion section (page 34, lines 129-142) to adressse the bioavailability issue,  gap between  preclinical and clinical studies,  challenges with extrapolation of results of preclinical experimental studies to dietary prevention in humans, and the value of alternative or co-adjuvant dietary prevention and treatment. 

Comment 2:

Several of the compounds reviewed in this manuscript seem to act on the Jak/STAT pathway indirectly (e.g. through Src or PI3K). How specific is the requirement for the STAT pathway for their anti-tumorigenic effect? We simply don't know. It is possible that these compounds affect multiple molecular pathways, and that Jak/STAT inhibition only partially contributes to their therapeutic potential. Therefore, the authors should more explicitly discuss that many of these natural compounds may act molecularly on as of yet unidentified targets and have indirect effects on several pathways that collaboratively contribute to malignancy. For instance, in the legends to Figure 2 and Figure 3, the authors say "... anticancer activities mediated through inhibition of (the) Jak/STAT pathway..." If the original research showed the appropriate molecular epistasis experiments to demonstrate that these compounds indeed depended on Jak/STAT inhibition to exert their anti-tumorigenic effect, then the authors should summarize the information accordingly. If, on the other hand, the original pre-clinical studies only showed a correlation between tumor growth suppression and Jak/STAT inhibition, then all statements indicating that a given compound acted *via* Jak/STAT inhibition should be revised to indicate that the anti-tumorigenic effect correlated with inhibition of the pathway.

Response:

The reviewer has made a terrific point. We completely agree with the reviewer and added a statement to acknowledge that the anticancer effects of the various phytochemicals as described in this review may be achieved through a variety of complementary and multi-targeted mechanisms, including modulation of yet unidentified targets, and one of these mechanisms could be the inhibition of the JAK/STAT signaling pathway (page 33, lines 122-125). Additionally, we have modified several sentences in table titles (Tables 1-3) and figure legends (Figures 2-5) to underscore that various phytoconstituents exerted their anticancer activity which can be correlated with inhibition of JAK/STAT signaling pathway.

Comment 3:

There is a strong emphasis throughout the manuscript on the therapeutic/medicinal effects of many natural compounds. It is perhaps stated most strongly in the Introduction (L62-66, and refs 8-14). Since the main premise of this review is to provide a potential molecular basis to the effectiveness of alternative medicines, the manuscript might benefit from a concise but more detailed description of the evidence of herbal medicines and dietary supplements with proven therapeutic or preventive effects on cancer treatment. In other words, if this review seeks to convey the idea that herbal medicine may function through its effect on known pathways related to cancer progression, the message would be supported by a more explicit description of the studies that provided sound evidence that herbal treatments have a demonstrable effect on the prevention or treatment of specific cancer types.

Response:

The objective of this review is to perform a comprehensive and crtical analysis of studies that demonstrate that various naturally-occurring phytoconstituents suppress the JAK/STAT signaling pathway as one of the key mechanisms of their cancer preventive and anticancer activities. We sincerely believe, a detailed description of the evidence of herbal medicines and dietary supplements with proven therapeutic or preventive effects on cancer treatment is outside the scope of our manuscript. Nevertheless, we have added text to discuss the value of natural products in the disciovery and development of anticancer drugs in the introduction section (page 2, lines 66-70).

Comment 4:

Section 3 (JAK/STAT signaling pathway) requires a thorough and careful revision. There is a lot of repetition (the pathway is described several times in different paragraphs), it often lacks logical order in the way that the pathway stages and components are described and it contains several phrases and terms that are either out of place or seem forcefully edited from other sources. For instance: "the improvement of the fetus" (L130), "receptor of the epidermal growth factor" (for Epidermal Growth Factor Receptor, or EGFR, L135), "the uplift (...) of cancer" or "chronic leukemia cells of the lymphoid" (for chronic lymphoblastic leukemia, L217). All of the previous seem forced attempts at changing the conventional terminology used to describe specific processes or molecules. While it is appreciated that the authors would want to re-phrase statements that they found in their review of the literature to avoid copying and pasting, they also need to be careful about respecting certain nomenclature conventions.

Response:

We greatly appreciate the reviewer’s careful observation. We have modified and delected the repeatative sentences on the mechanism (page 3, section 3: JAK/STAT signaling pathway). We have tried to incorporate a logical order for better understanding of the whole mechanism. Additionally, we have also corrected those conventional terminology. All modifications are highlighted in red color front.

Comment 5:

There are many cell lines that are mentioned by name throughout the article, but there's no mention to what type of cell line they are. Some examples include A172 and U373MG (L20, p13), U266 (L29, p13), THP1 and HEL (L184, p16) and others. I would recommend that the authors peruse their manuscript carefully to make sure that they indicate the cell types associated with cell lines mentioned, for those who are not experts and familiar with a specific cell line.

Response:

We totally agree with the reviewer. Along with the cell line names, we have provided the information on source (organ/tissue) or cancer type in the text as well as tables.

Comment 6:

L90, p32: "Since natural components...cancer treatment" I beg to differ with the authors on this statement on two levels: 1) if natural components may not have adverse most often; but they usually do: we call them natural poisonous plants; 2) natural compounds present in low amounts in natural sources may not have noxious effects. But they can turn very toxic when extracted, purified and concentrated for use in therapy. For instance: taxol. If what the authors meant is that the consumption of natural herbs, plants and fungi that contain active phytocompounds will be rarely toxic, then it should be made more clearly explicit. Natural components that are given at higher (therapeutic) doses very often cause adverse side effects. I would therefore invite the authors to revise this sentence.

Response:

We are in absolute agreement with the reviewer and accordingly have modified this section as suggested (page 32, lines 86-93).

Minor points

Comment 1:

L41-43: What do the authors mean by cancers being "dominated" by viral or bacterial infection and genetic mutations? What else could cause cancer?

Response:

We have modified the sentence as “Tumors associated with viral or bacterial infection and genetic mutations are known to influence the cancer growth rates” (page 1, lines 41 and 42). Another sentence in the same paragraph has also been revised as “Various risk factors, such as smoking, chewing tobacco, alcohol, obesity, chronic inflammation, age, ethnicity and geographical location, are the major determinants for developing cancer (page 1, line 44 to page 3, line 46).

Comment 2:

L56-57: The authors should consider changing "...perceived as potential mechanisms of cancer growth..." to "...potential approaches to cancer treatment..." It makes better sense in the context of targeting and inhibiting the hallmarks of cancer progression.

Response:

We have revised the sentence as suggested (page 2, line 57).

Comment 3:

L67-77: This paragraph is poorly organized - it leads with the potential mechanisms by which bioactive phytocompounds might impact cancer progression or treatment, and ends with broader statements about their potential. Perhaps the order of statements should be revised/reversed.

Response:

We have revised the order of statements as suggested (page 2, lines 71-79).

Comment 4:

L78: consider "...initiating CHANGES in gene expression..."

Response:

The recommendation has been incorporated (page 2, line 82).

Comment 4:

L83: This sentence is confusing. Are the authors implying that "Inhibiting the JAK/STAT signaling pathway may [suppress tumorigenesis] by preventing [apoptosis]"? If inhibiting JAK/STAT is anti-apoptotic, then inhibiting JAK/STAT would be PRO-tumorigenic. This sentence needs revising.

Response:

We acknowledge that there has been a problem with the sentence construction. We have composed two sentences for better presentation (page 2, lines 87-90).

Comment 5:

L108: What is meant by "...STAT retained residues..."? Please clarify/revise.

Response:

This sentence has been revised in order to promote clarity (page 3, lines 114-116).

Comment 6:

L110-111: Consider "...DNA and stimulates THE EXPRESSION OF GENES that are responsive to cytokines..."

Response:

The recommended modification has been made  (page 3, line 117).

Comment 7:

L120: Nonsensical sentence - please revise.

Response:

The sentence has been modified (page 3, lines 126-128).

Comment 8:

L144 (Fig 1 legend): Consider "...activation leading to up regulation of survival and tumor cell proliferation GENES..."

Response:

We have revised the sentence (page 4, lines 146 and 147).

Comment 9:

L155-159: Refs 48 and 49 do not seem to address the inhibition of inflammatory pathways by natural compounds (at leas by title) - please correct, or clarify further.

Response:

We have cite an appropriate reference (ref. 48) for the sentence “Since then, natural components…..inflammatory pathway (page 4, line 159). The other reference numbers are adjusted in this paragraph.

Comment 10:

L204-205: "...has identified domains or residues whole functions to be delineated..." Is this a broken sentence? Please revise/edit.

Response:

This sentence has been modified (page 5, lines 206-208).

Comment 11:

L208-209: The meaning of "The canonical signaling is a characteristic of tyrosine phosphorylated STAT functioning that serve as transcription factors" is implicit, but the phrase needs rewriting.

Response:

We have modified the sentence (page 5, lines 210 and 211).

Comment 12:

L31, p13: delete "and" in "...by siRNA and abrogated..."

Response:

The necessary correction has been made (page 13, line 31).

Comment 13:

L66, p14: "various" should be changed to "two".

Response:

The term “various” has been replaced with “two” (page 14, line 66).

Comment 14:

L91-93, p14: I find this statement confusing: a) How are the IFN-a and JAK/STAT pathways related to each other again? (the authors could summarize in 1-2 sentences what is known about their connection); b) if emodin inhibits STAT3 phosphorylation, would it act by "stimulating" (as written) or by INHIBITING the JAK/STAT signaling pathway. It's confusing.

Response:

a) We have revised the sentence as “Emodin stimulated the antiproliferative role of interferon α/β (IFN-α/β) by increasing the STAT1 phosphorylation, diminishing the STAT3 phosphorylation and expanded the expression of an endogenous gene activated by IFN-α” (page 14, lines 86-88).

b) This sentence has been modified to “…emodin enhanced the antiproliferative action of IFN-α/β by inhibition of JAK/STAT signaling pathway through restraining 26S proteasome-stimulated IFNAR1 degradation” (page 14, lines 92-94).

Comment 15:

L85-87, p32: "This section...can be drawn" looks like boilerplate language provided by MDPI as guidance for the authors. It should be taken out.

Response:

We sincerely apologize for this inadvertent mistake and have deleted the redundant sentence.

On behalf of my co-authors, I once again express my sincere thanks to the erudite reviewer for the valuable suggestions and constructive input to improve the quality of our manuscript.